# Small molecule inhibition of cGAS reduces interferon expression in primary macrophages from autoimmune mice

Jessica Vincent[1], Carolina Adura[2], Pu Gao[3,4], Antonio Luz[2,7], Lodoe Lama[2,5], Yasutomi Asano[6], Rei Okamoto[6], Toshihiro Imaeda[6], Jumpei Aida[6], Katherine Rothamel[1], Tasos Gogakos[2,5], Joshua Steinberg[2,5], Seth Reasoner[1], Kazuyoshi Aso[6], Thomas Tuschl[2,5], Dinshaw J. Patel[3], J. Fraser Glickman ⓘD [2] & Manuel Ascano ⓘD [1]

Cyclic GMP-AMP synthase is essential for innate immunity against infection and cellular damage, serving as a sensor of DNA from pathogens or mislocalized self-DNA. Upon binding double-stranded DNA, cyclic GMP-AMP synthase synthesizes a cyclic dinucleotide that initiates an inflammatory cellular response. Mouse studies that recapitulate causative mutations in the autoimmune disease Aicardi-Goutières syndrome demonstrate that ablating the cyclic GMP-AMP synthase gene abolishes the deleterious phenotype. Here, we report the discovery of a class of cyclic GMP-AMP synthase inhibitors identified by a high-throughput screen. These compounds possess defined structure-activity relationships and we present crystal structures of cyclic GMP-AMP synthase, double-stranded DNA, and inhibitors within the enzymatic active site. We find that a chemically improved member, RU.521, is active and selective in cellular assays of cyclic GMP-AMP synthase-mediated signaling and reduces constitutive expression of interferon in macrophages from a mouse model of Aicardi-Goutières syndrome. RU.521 will be useful toward understanding the biological roles of cyclic GMP-AMP synthase and can serve as a molecular scaffold for development of future autoimmune therapies.

[1] Vanderbilt University School of Medicine, Nashville, TN 37027, USA. [2] The Rockefeller University, New York, NY 10065, USA. [3] Structural Biology Program, Memorial Sloan-Kettering Cancer Center, New York, NY 10065, USA. [4] Key Laboratory of Infection and Immunity, CAS Center for Excellence in Biomacromolecules, Institute of Biophysics, Chinese Academy of Sciences, Beijing 100101, China. [5] Howard Hughes Medical Institute Laboratory for RNA Molecular Biology, New York, NY 10065, USA. [6] Tri-Institutional Therapeutics Discovery Institute, New York, NY 10021, USA. [7] Present address: Regeneron Pharmaceuticals Incorporated, Tarrytown, NY 10591, USA. Jessica Vincent, Carolina Adura, and Pu Gao contributed equally to this work. Correspondence and requests for materials should be addressed to D.J.P. (email: pateld@mskcc.org) or to J.F.G. (email: fglickman@mail.rockefeller.edu) or to M.A. (email: manuel.ascano@vanderbilt.edu)

The innate immune system contains protein sensors to detect aberrantly modified and/or mislocalized nucleic acids, perceiving such molecules as foreign or markers of cellular stress[1–5]. Sensing of aberrant nucleic acids leads to the activation of downstream signal transduction pathways and inflammatory responses through the upregulation of type I interferon genes. One such pattern recognition receptor is cyclic GMP-AMP synthase (cGAS, official gene name MB21D1)[6, 7], which detects cytoplasmic double-stranded DNA (dsDNA), indicative of an infection by a virus or bacterial pathogen or mislocalization of nuclear or mitochondrial DNA[8–10]. Upon binding to dsDNA, cGAS utilizes ATP and GTP to synthesize the only known metazoan cyclic-dinucleotide, cyclic GMP-AMP (cGAMP or c[G(2′,5′)pA(3′,5′)p])[11–16], a molecule in which guanosine and adenosine are linked with a 2′,5′- and a 3′,5′-phosphodiester bond following sequential ligation at the same active site. cGAMP acts as a second messenger, diffusing and binding to the endoplasmic reticulum membrane-bound adapter protein, Stimulator of interferon genes (STING), thus

initiating a signal transduction cascade which leads to the activation of the transcription factor interferon regulatory factor 3 (IRF3) and the upregulation of cytokines including type I interferon beta 1 (IFNB1)[11, 17–21]. Many studies have now implicated the importance of cGAS in the innate immune response to intracellular and prokaryotic pathogens such as *L. monocytogenes*[22], *C. trachomatis*[23], *L. pneumophila*[24], and *M. tuberculosis*[24–26]. Moreover, the cGAS pathway can detect or be suppressed by viruses during infection, including members of the herpes family[27, 28], the oncogenic viruses hAD5 and HPV18[29], as well as retroviruses including HIV[30], highlighting its role as a key sentinel against pathogenic infection.

While the cellular immune response to dsDNA plays an indispensable role in pathogen defense, an abnormal response to dsDNA has been shown to be an important factor in the etiology of hyper-inflammatory or autoimmune disorders. A hallmark of autoimmune disorders such as systemic lupus erythematosus (SLE) is the development of patient serum immunoreactivity against dsDNA[31]. Inactivating mutations in the enzyme

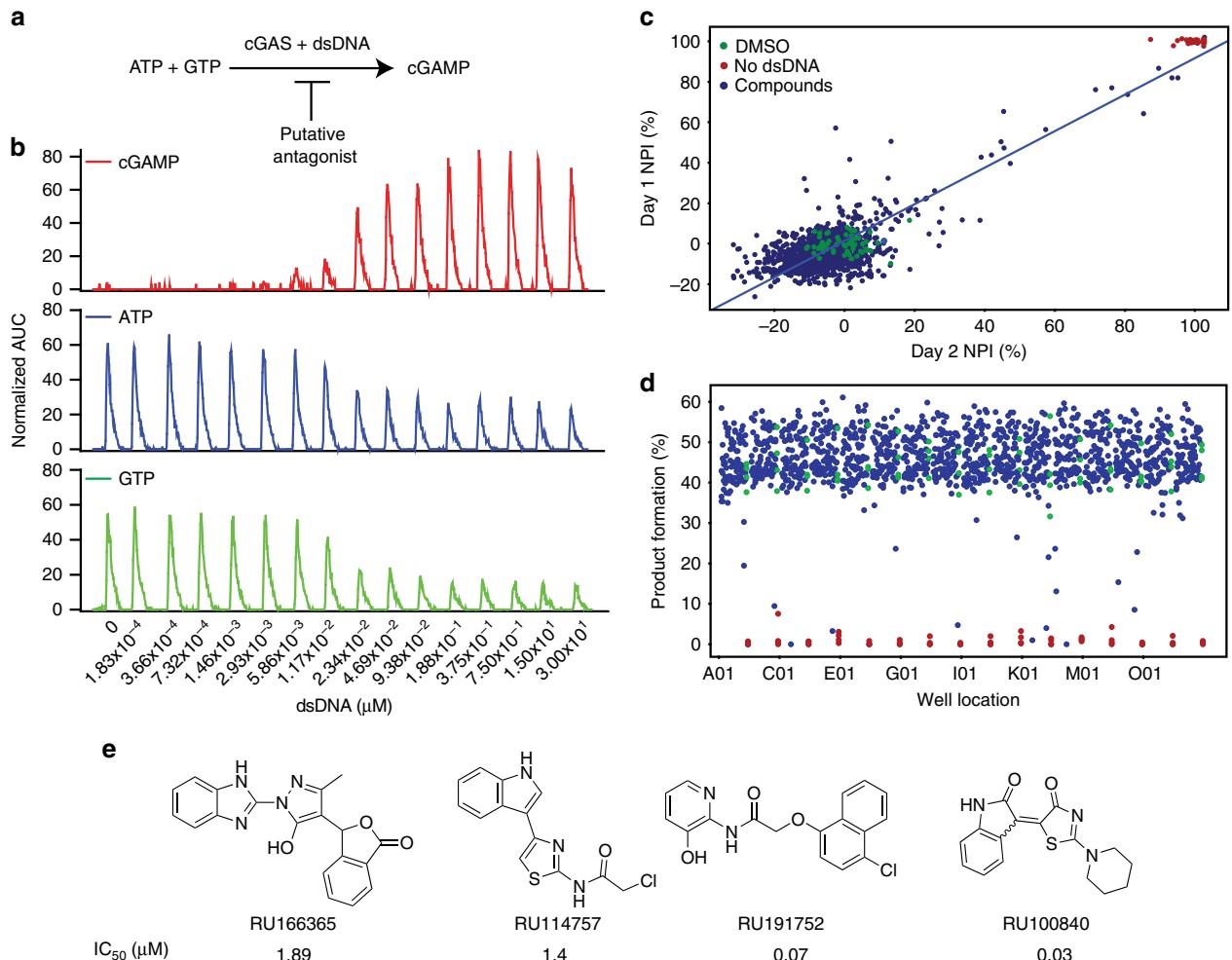

**Fig. 1** Development of a high-throughput screen for the identification of cGAS inhibitors. **a** Schematic of immune stimulatory dsDNA-dependent cGAS synthesis of cyclic GMP-AMP (*cGAMP*). **b** The enzymatic activity of cGAS is determined by monitoring the consumption of ATP (*blue*) and GTP (*green*), and the generation of cGAMP (*red*) using an RF-MS. The assay was incubated for 120 min using 60 nM cGAS. Normalized extracted ion count values are plotted. **c** For the high-throughput screen, 300 nM dsDNA was used to stimulate mouse cGAS activity; % cGAMP product formation was measured against 1266 compounds (*blue dots*) at a final concentration of 12.5 µM using Sigma-Aldrich LOPAC (library of pharmacologically active compound) library plates. The results of two independent days, as technical replicates, are plotted against each other as normalized percent of inhibition (NPI). The negative control DMSO is shown in *green* and the positive control (no dsDNA) is shown in *red*. Data were analyzed using Vortex software and the coefficient of correlation was 0.86. **d** The results using the LOPAC library resulted in the assay having a *Z* prime factor of 0.76. **e** Following the high-throughput screen from over 100,000 small-molecule compounds, the following four were selected for additional characterization. See text for details on the triage process

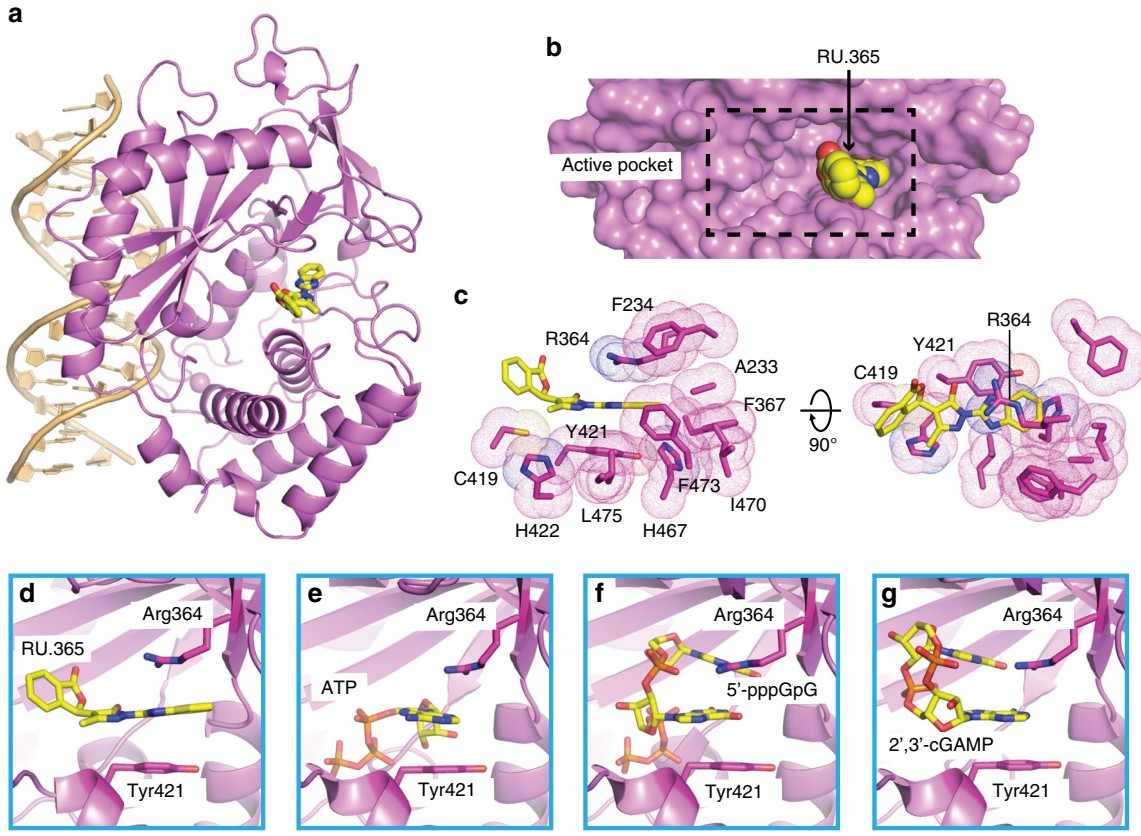

**Fig. 2** Ternary crystal structure of cGAS with dsDNA and RU.365. **a** Overall structure of the cGAS-dDNA-RU.365 complex, with cGAS in *violet*, dsDNA in *light orange*, and ligand in *yellow*. RU.365 is shown in *stick* representation. **b** Close-up of the cGAS-binding pocket (shown in surface representation and *violet*) and bound RU.365 (show in space-filling representation and *yellow*). The *black dashed box* indicates the entire binding pocket of cGAS. **c** The amino acids (shown in *dots* and *stick*, in *violet*) surrounding the bound RU.365 (shown in *stick*, in *yellow*). **d–g** The conserved stacking interaction mediated by Arg364 and Tyr421 with the bound RU.365 (**d**), ATP (**e**), 5′-pppGpG (**f**), and 2′,3′-cGAMP (**g**). The bound ligands and these two residues are shown in *stick* representation

responsible for degrading cytoplasmic DNA, the human DNA exonuclease TREX1, cause Aicardi-Goutières syndrome[32] (AGS) or Chilblain lupus[33], both autoimmune disorders manifesting overproduction of interferon and whose clinical symptoms overlap with SLE. Knockout studies using bone marrow-derived macrophages (BMDMs) of Trex1 null mice have shown elevated levels of interferon, which are normalized upon knockout of cGAS[34, 35], Sting[36, 37], or IRF3[36], substantiating the contribution of cGAS function in these diseases. Moreover, recent work has implicated mitochondria as a cell-intrinsic source of aberrantly localized DNA, indicating that mitochondrial stress can promote an inflammatory response by triggering cGAS activity[38, 39]. Since damaged mitochondria can be the consequence of many kinds of cellular insults, the number of inflammatory responses for which cGAS is a major driver may still be underestimated[40].

Given its central importance in innate immunity, small-molecule inhibitors of the enzymatic activity of cGAS could be used to enhance research efforts to dissect cGAS mechanism and regulation of innate immunity. Additionally, small molecules that target cGAS can be an important step toward the development of drugs in the treatment of human disorders relevant to a dysregulated cGAS pathway. Recently, an in silico screen has identified existing anti-malarial small molecules with some efficacy toward inhibiting cGAS by interfering with its association with dsDNA[40]. However, anti-malarial drugs can intercalate dsDNA, and their mechanism of inhibition appears to rely on non-specific interactions with nucleic acid rather than direct association and interference of cGAS and its enzymatic activity. Intercalating compounds have a higher propensity of having side effects in the cell.

Here, we report the discovery and characterization of a class of compounds that binds to the catalytic pocket of cGAS and inhibits its dsDNA-stimulated activity. We define structure-activity relationships and present the crystal structure of an inhibitor–enzyme–dsDNA. We further demonstrate the potency and selectivity of a chemically improved inhibitor, RU.521, in cellular assays showing that while it inhibits cGAS-mediated interferon upregulation, it has reduced to no effect on inflammatory pathways independent of cGAS. Furthermore, RU.521 suppresses the chronically elevated levels of type I interferon observed in primary macrophages from Trex1 null mice, a model of AGS. We expect this chemical scaffold to facilitate studies of the physiological roles played by cGAS, which will foster a greater understanding of innate immune mechanisms.

## Results

**Discovery of a small-molecule inhibitor of cGAS.** We sought to develop a high-throughput screen using a small-molecule library to identify compounds that could modulate enzymatic production of cGAMP. In this report, we describe our discovery and characterization of cGAS inhibitors. The enzymatic synthesis of the cyclic dinucleotide cGAMP is readily performed in cell-free conditions by combining purified recombinant mouse cGAS enzyme with pre-annealed 45-bp dsDNA in a reaction buffer, and adding ATP and GTP as substrates (Fig. 1a). We used an Agilent RapidFire mass spectrometry system (RF-MS) to simultaneously quantify the abundance of substrates and product in reaction

**Table 1 X-ray statistics for cGAS-DNA-inhibitor complexes**

| | cGAS-dsDNA-RU.365 | cGAS-dsDNA-RU.332 | cGAS-dsDNA-RU.521 |
|---|---|---|---|
| *Data collection* | | | |
| Space group | I222 | I222 | I222 |
| Cell dimensions | | | |
| $a, b, c$ (Å) | 85.5, 98.8, 130.2 | 85.3, 98.6, 129.2 | 85.6, 98.4, 130.4 |
| $\alpha, \beta, \gamma$ (°) | 90.0, 90.0, 90.0 | 90.0, 90.0, 90.0 | 90.0, 90.0, 90.0 |
| Resolution (Å) | 50–2.13 (2.19–2.13)[a] | 50–2.18 (2.24–2.18)[a] | 50–1.83 (1.87–1.83)[a] |
| $R_{sym}$ or $R_{merge}$ | 0.053 (1.544) | 0.073 (1.363) | 0.044 (1.520) |
| $I/\sigma I$ | 13.3 (1.1) | 12.8 (1.3) | 17.7 (1.1) |
| Completeness (%) | 98.8 (99.7) | 99.0 (95.1) | 98.4 (94.1) |
| Redundancy | 3.2 (3.3) | 3.6 (3.4) | 4.3 (4.3) |
| *Refinement* | | | |
| Resolution (Å) | 50–2.13 (2.19–2.13)[a] | 50–2.18 (2.24–2.18)[a] | 50–1.83 (1.87–1.83)[a] |
| No. of reflections | 30,303 | 28,373 | 45,239 |
| $R_{work}/R_{free}$ | 0.213/0.248 | 0.242/0.281 | 0.194/0.232 |
| No. of atoms | | | |
| Protein | 3529 | 3529 | 3529 |
| Ligand/ion | 2 | 2 | 2 |
| Water | 66 | 82 | 180 |
| *B*-factors | | | |
| Protein | 72.44 | 67.63 | 58.26 |
| Ligand/ion | 156.50 | 86.72 | 80.73 |
| Water | 56.46 | 53.97 | 49.90 |
| R.m.s. deviations | | | |
| Bond lengths (Å) | 0.009 | 0.009 | 0.009 |
| Bond angles (°) | 1.228 | 1.253 | 1.153 |

[a]One crystal for each structure; values in parentheses are for highest-resolution shell

mixtures containing recombinant cGAS and dsDNA. Figure 1b shows a typical RF-MS chromatogram where ATP and GTP were present at equimolar concentration (0.3 mM), while the dsDNA concentration was varied between 0 to 3 µM. After 120 min, reactions were stopped and the abundance of ATP, GTP, and cGAMP were measured. The $EC_{50}$ for dsDNA is 0.017 µM (Supplementary Fig. 1a). At 0.3 µM dsDNA, the rate of product formation was linear over 120 min at room temperature yielding approximately 50% of cGAMP (Supplementary Fig. 1b). We then measured under steady-state conditions the effect of substrate concentration on catalytic rate. We found that the saturation curves consistently and significantly fit better to a sigmoidal curve rather than a hyperbolic function. This suggests that the substrate binding works cooperatively, and is in agreement with the structural findings that show cGAS can exist as a 2:2 dimer with two enzymes binding to two DNA molecules[41]. Thus, it is possible that initial binding of ATP and GTP to the active site of a cGAS molecule, could cause a higher affinity binding of substrates to occur with the second cGAS molecule. We used an allosteric sigmoidal fitting and found that ATP has a $K_m$ of $86 \pm 6.7$ and a $k_{cat}$ 23 (min$^{-1}$) The $K_m$ and $k_{cat}$ for GTP are $96.4 \pm 5.1$ µM and 23 (min$^{-1}$) respectively (Supplementary Fig. 1c, d). Taken together, these results show the utility of RF-MS for measuring in vitro cGAS activity and that reaction conditions were amenable to high-throughput screening.

We also tested whether human cGAS could be used in the high-throughput screen by performing an enzyme progress curve (Supplementary Fig. 2). The signal for human cGAS is undetectable under the same conditions used with mouse cGAS. Due to the low signal, human cGAS was not suitable for accurate kinetic characterization using the technique presented in this paper and the screen and subsequent validation assays were performed with the murine version.

A pilot study was conducted in two different days using 1268 compounds from the Sigma Aldrich LOPAC compound collection in order to test the statistical robustness of the assay.

We obtained a linear regression coefficient of 0.86 (Fig. 1c) and a $Z'$ value of 0.76 (Fig. 1d). Subsequently, a total of 123,306 compounds were screened (Supplementary Table 1) at a concentration of 12.5 µM. A cutoff of 60% inhibition normalized by the intra-plate controls and a $Z'$ criterion of > 0.5 was applied yielding 229 (0.19%) compounds. These were re-tested in triplicate concentration response experiments. One-hundred and four compounds showed an $IC_{50} \leq 10$ µM. Of these, 17 contained structural motifs that are recurrent in PAINS[42] and, therefore, were removed from further consideration, yielding 87 compounds. Further high-performance liquid chromatography (HPLC) mass spectrometry analysis determined that 49 compounds showed the correct molecular mass and had a purity $\geq 85\%$ in the first chromatography gradient run in positive ion mode (Supplementary Table 2). These 49 compounds were evaluated for potential DNA intercalation by measuring the displacement of acridine orange bound to dsDNA by means of fluorescence polarization (FP). Fourteen compounds showed > 50% of acridine orange displacement from dsDNA compared to 30 µM mitoxantrone—a known DNA intercalator[43]. From 35 remaining compounds, 4 compounds (Fig. 1e) had 50% inhibitory concentrations ($IC_{50}$'s) ranging from 0.03 µM to 1.9 µM (confirmed with independent lots of compound) and were then prioritized based on stability, potency, and structural diversity for further testing in crystallization trials.

**RU.365 binds to the active site of cGAS.** We were able to solve the crystal structure of RU166365 (hereafter as RU.365) in complex with cGAS and dsDNA (Fig. 2a and Supplementary Fig. 4), allowing us to better examine the structural basis for the affinity of this inhibitor. The ternary complex was obtained by using the co-crystallization method and the structure was solved at 2.13 Å resolution by molecular replacement based on PDB: 4K96 (X-ray statistics in Table 1). The final cGAS-dsDNA-RU.365 ternary complex shows good refinement statistics

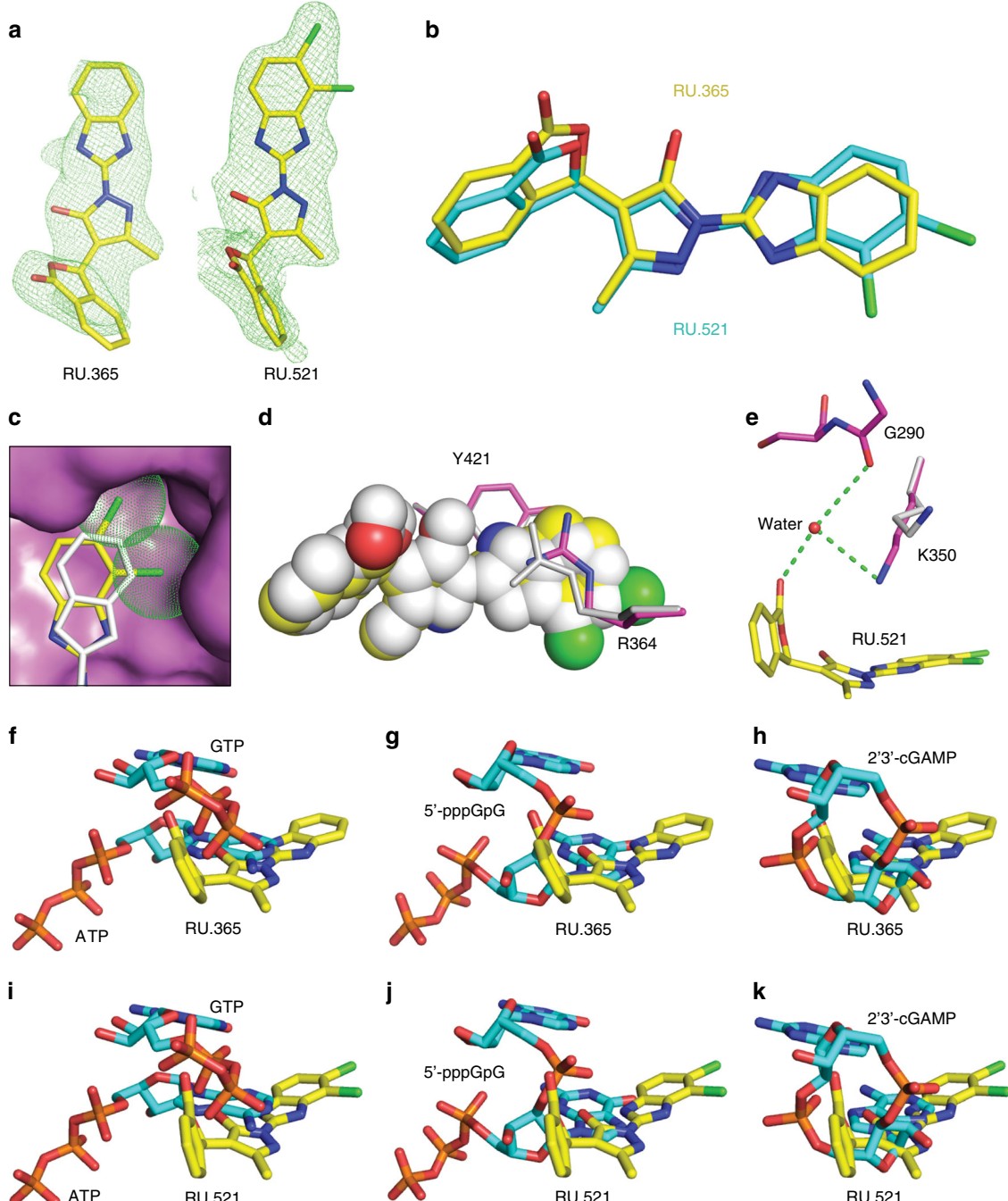

**Fig. 3** A structural comparison of RU.365 with its derivative RU.521. **a** Omit Fo-Fc electron density (*green mesh*) of RU.365 and RU.521 contoured at 3σ. **b** Superposition of structures of bound RU.365 (*yellow*) and RU.521 (*cyan*). **c** The dichloro substitution target deeper in the cGAS pocket. RU.521 (*yellow*) and RU.365 (*silver*) are shown in *stick*, with two chlorines shown in *dots*. cGAS (*violet*) is shown in *surface view*. **d** The stacking interaction between RU.521 (*yellow*)/RU.365 (*silver*) and R364/Tyr421. **e** Water-mediated interaction between RU.521 (*yellow*) and cGAS (*violet*). Lys350 in RU.365 complex is shown in *silver*. Superposition of structures of bound RU.365 **f**–**h** and RU.521 **i**–**k** with those of substrates, intermediate, and product. All the ligands are shown in *stick* representation. Inhibitors are shown in *yellow*. Substrates/Intermediate/Product are shown in *cyan*

($R_{work}$: 21.3%; $R_{free}$: 24.8%) and a ligand geometry well-defined by electron density (Fig. 3a). cGAS adopts an active conformation in this ternary complex characterized by a DNA-induced "open pocket" (Fig. 2a, b), similar to our previously reported structure of the complex of cGAS with bound dsDNA and cGAMP[14]. RU.365 occupies only one side of the cGAS pocket, as does cGAMP in our previously reported structure[14], leaving the remainder of the pocket filled with solvent (Fig. 2b). Of the amino acids surrounding RU.365, the aromatic ones are Phe234, Phe367,

Tyr421, His422, His467, and Phe473, the hydrophobic amino acids are Ala233, Ile470, and Leu475), and the charged or polar amino acids are Arg364 and Cys419 (Fig. 2c). The benzimidazole ring and a part of the pyrazole ring of bound RU.365 partially stack between the guanidinium group of Arg364 and Tyr421 (Fig. 2c, d), which represent the key intermolecular contacts between RU.365 and cGAS. Of note, point mutations of Arg364 and Tyr421 render it incapable of responding to dsDNA, and no interferon upregulation is observed[14]. Stacking interactions

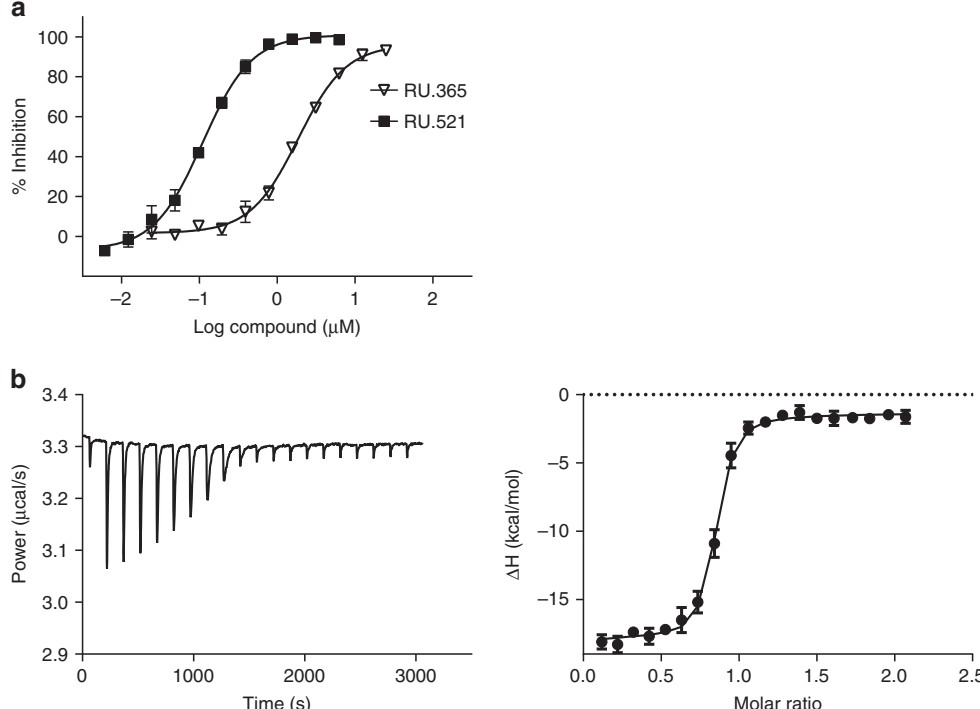

**Fig. 4** cGAS inhibition by RU.365 and its chemical active analogs. **a** In vitro concentration response curves for RU.365 and the most potent analog synthesized in-house, RU.521, in the presence of dsDNA, ATP and GTP as measured by RF-MS. The IC50 values for RU.365, RU.521 and less potent analogs are listed in Supplementary Table 4. Values are averages of triplicate determinations with SD indicated. **b** Isothermal titration calorimetry-binding curves for RU.521 titrated into cGAS/dsDNA complex

were also found in substrate- (ATP, Fig. 2e), intermediate- (5′-pppGpG, Fig. 2f), and product-bound (cGAMP, Fig. 2g) structures[14], indicating a conserved ligand-binding strategy of cGAS. The angle between the pyrazole ring and phthalide ring of bound RU.365 is approximately 120°, which is mainly caused by the interaction between Cys419 and the hinge region of these two rings. The remaining parts of the phthalide ring face the solvent and form no interactions with cGAS. Surprisingly, there is no hydrogen bonding interaction between the inhibitor and the protein, despite the presence of four nitrogen and three oxygen atoms in RU.365.

**RU.365 analogs bind to the active site of cGAS**. We also solved the crystal structure of RU281332 (hereafter as RU.332), an analog of RU.365, in complex with cGAS and dsDNA at 2.18 Å resolution (Supplementary Figs. 3 and 4; X-ray statistics in Table 1). The only difference between RU.365 and RU.332 is the substitution of the benzimidazole ring with a benzothiazole ring; the two bound inhibitors can be superimposed very well with each other (Supplementary Fig. 3a, b). We can distinguish S from N of the benzothiazole ring by checking the Fo-Fc map during refinement. The sulfane group is facing the aromatic/hydrophobic surface of the binding pocket, while the remaining methanamine group is pointing into the solvent region. The interactions between RU.332 and cGAS are very similar to that of RU.365.

To improve the binding affinity and inhibitory activity of the lead compound, we set-up a structure-guided chemical synthesis process. Among many synthesized RU.365 derivatives, the compound RU320521 (hereafter RU.521) with a dichloro substitution (Fig. 3a) showed the best inhibition activity (see below). We determined the crystal structure of cGAS-dsDNA-RU.521 ternary complex at 1.83 Å resolution by using a co-crystallization approach (X-ray statistics in Table 1). The two modified chlorines in RU.521 target deeper in the pocket

of cGAS and subsequently push the benzimidazole ring away from the protein (Fig. 3b, c), which in turn increases the stacking surface of RU.521 with Arg364 and Tyr421 (Fig. 3d). In addition, we observed a water-mediated interaction between the aldehyde group of the phthalide ring and the main-chain of Gly290 and the side-chain of Lys350 (Fig. 3e), which is absent in the RU.365 structure. The above structural features of RU.521 make it a more potent inhibitor than RU.365.

To our surprise, by superposing the structures of inhibitor complexes with the structure of cGAS in free state, we found that RU.365/RU.332/RU.521 could also fit the "smaller active site" of apo-cGAS in the dsDNA-free state without notable steric effects (Supplementary Fig. 5). To confirm this we used isothermal titration calorimetry (ITC) to measure the affinity of RU.365 in the absence or presence of dsDNA. There was not a large difference between the affinity of RU.365 with and without dsDNA, both being between 64.1 nM and 98.5 nM, respectively (Supplementary Fig. 5).

**A structural comparison of cGAS ternary complexes**. To investigate how RU.365/RU.332/RU.521 might inhibit cGAS activity, we compared the inhibitor complexes to cGAS-DNA in complex with substrate, intermediate, and product. When comparing the bound inhibitors with the substrates (ATP/GTP)[14, 16], we observed heavy steric clashes between inhibitors with both ATP and GTP. The benzimidazole-pyrazole and phthalide moieties of RU.365 overlap with the adenine ring of ATP and the α/β-phosphates of GTP, respectively (Fig. 3f). The inhibitors also show overlapped occupancy with the bound covalent intermediate analog 5′-pppG(2,5)pG[14], with the benzimidazole-pyrazole moiety clashing against the guanine ring of pppG and the phthalide ring against the sugar ring of pppG, as well as the linker phosphate (Fig. 3g). Compared to the substrates and intermediate, the bound cGAMP product[14] shows better

superimposition with RU.365 (Fig. 3h). The entire RU.365 molecule overlaps well with the AMP moiety of cGAMP, with the benzimidazole ring of RU.365 inserting deeper into the pocket than the adenine ring of cGAMP. RU.332 (Supplementary Fig. 3c–e) and RU.521 (Fig. 3i–k) show similar superposition results to that of RU.365. The above comparison suggests that RU.365/RU.332/RU.521 could potentially inhibit every catalytic step (substrate binding, intermediate and product formation) of cGAS, or could displace substrates from the catalytic pocket.

To further explore this, we performed ITC-binding experiments titrating RU.365 into the complex cGAS/dsDNA in the presence or absence of ATP or GTP. We found that the affinity of RU.365 decreased in presence of GTP or ATP (64 nM in absence of GTP vs. 298.2 nM, in the presence of GTP, 104 nM $K_d$ in presence of ATP, Supplementary Fig. 6). Taken together, RU-365 appears to impede the binding of both GTP and ATP to the enzyme active site. To determine whether RU.365 showed a competitive mechanism of inhibition enzyme kinetics, experiments were performed to examine the effect of inhibitor concentration on $V_{max}$ and $K_m$. We found that $K_m^{app}$ of both substrates did not increase in the presence of increasing concentrations of RU.365. Under the same conditions the $V_{max}$ varied significantly (Supplementary Fig. 7).

**Improving RU.365 potency through structure-activity studies.** To examine the structure-activity relationship of cGAS inhibitors and seek molecules with improved potency, we tested in-house synthesized and commercially available analogs of RU.365 harboring substitutions of the atoms in the benzimidazole, the isobenzofuran-3-one, or the central pyrazol-5-ol ring systems. We performed concentration-response experiments in the RF-MS assay and derived 50% inhibitory concentration ($IC_{50}$) values for each compound. We tested analogs with targeted substitutions in groups that can function as H-bond donors or acceptors since we previously observed that replacing the 1-N benzimidazole's nitrogen by sulfur (RU.332, Supplementary Table 3) did not change the potency of cGAS inhibition, whereas ring conversion from isobenzofuranone to isoindolinone (RU320465) abolished inhibitory activity (Supplementary Table 4). Furthermore,

modifications of the isobenzofuran-3-one group showed only minor variations in potency (RU320469 and RU320468, Supplementary Table 3).

Importantly, and consistent with our structural model, modifications that increased the hydrophobic character of the benzimidazole improved potency, suggesting an interaction of this moiety with a hydrophobic environment in the target protein. Substantiating this hypothesis, we found that compounds in which aromatic hydrogens of the benzimidazole were replaced by halogens or methyl groups showed increased inhibitory potency (RU.320519, RU320461, RU320462, RU320520, RU320467, RU320582, and RU.521, Supplementary Table 3). A 17-fold improvement in potency was observed in analogs with 6,7- or 5-7-dichloro substitution in compounds RU.521 and RU320582. Additional analogs and their potencies are shown in Supplementary Table 4. Concentration response curves of RU.365 and its most potent analog RU.521 are shown in Fig. 4a. ITC experiments were used to confirm the binding of RU.521 to the cGAS/dsDNA complex ($K_d = 36.2$ nM, Fig. 4b and Table 2).

**Inhibition of dsDNA-induced signaling in macrophage cells.** We next evaluated the activity of the in vitro validated compounds in cellular assays to determine their efficacy in suppressing dsDNA-dependent activation of cGAS and subsequent upregulation of type I interferon genes. Introduction of dsDNA into RAW macrophage cells leads to marked upregulation of type-I interferons through activation of IRF3 in a cGAS and Sting dependent manner[6, 11–14, 21]. We used RAW cells that stably express an interferon-responsive element coupled to a luciferase gene (IFNB1-Luc), enabling us to utilize luciferase signal as the readout of cGAMP production. We performed a series of cellular concentration-response analyses to determine the $IC_{50}$ values of the small molecules in dsDNA-activated RAW cells. We observed that the compounds exhibited $IC_{50}$ values ranging from 700 nM to 13.60 µM. The cellular $IC_{50}$ value for RU.365 (4.98 µM, Fig. 5a) was approximately twofold lower than RU.332 (13.60 µM, Fig. 5b). The derivatized analog, RU.521, had an $IC_{50}$ of 700 nM (Fig. 5c) and was the best at suppressing dsDNA-activated reporter activity.

**Selective inhibition of cGAS-mediated signaling by RU.521.** To evaluate whether RU.521 and its analogs might affect other innate immune signaling pathways beyond dsDNA activation, we stimulated RAW macrophage cells with a selection of other immunogenic ligands. Specifically, we exposed cells to ligands for RIG-I (5′ppp-HP20 RNA[44]), Tlr2/1 (Pam3CSK4), Tlr3 (poly(I:C)), Tlr4 (lipopolysaccharide, LPS), and JAK/STAT signaling (recombinant Ifnb) in the presence or absence of

**Table 2 Dissociation constant and thermodynamic parameters of RU.521**

|  | rM | $K_d$ (nM) | $\Delta H$ (cal/mol) | $T^\star\Delta S$ (cal/mol) | $\Delta G$ (cal/mol) |
|---|---|---|---|---|---|
| RU.521 | 0.8071 | 36.17 | −16290 | −6134 | −10150 |

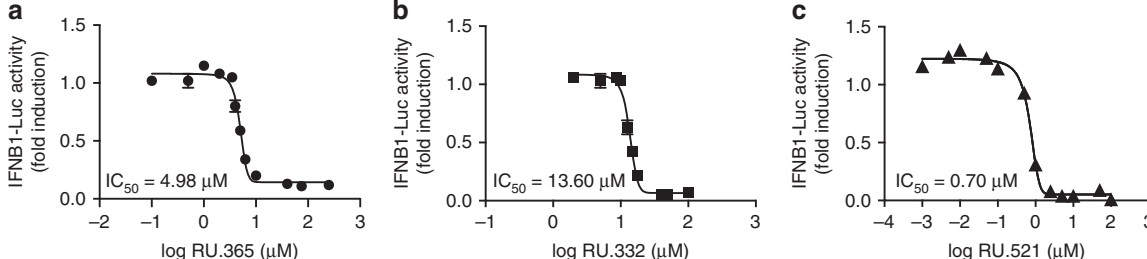

**Fig. 5** Small-molecule inhibition of cGAS-dependent interferon induction in cellular assays. RU.365, RU.332, and RU.521 were tested in cellular assays for their effectiveness in inhibiting cGAS activity in dsDNA-stimulated RAW macrophages. The inhibition of type I IFN response via dsDNA was monitored via an interferon sensitive promoter coupled to a luciferase gene. The concentration response curves for RU.365 (**a**), RU.332 (**b**), and RU.521 (**c**) in RAW cells are shown. *Error bars* represent SEM

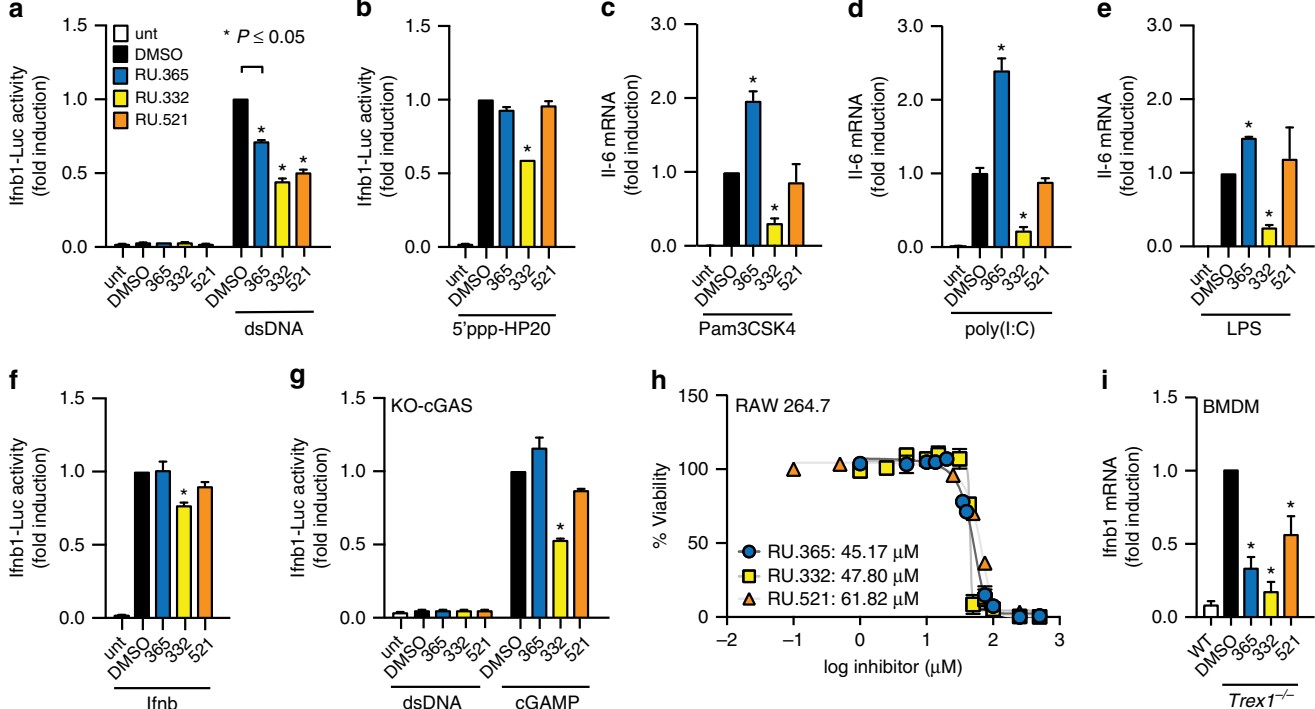

**Fig. 6** Potent and selective inhibition of cGAS activity in RAW macrophage and BMDM cells from an Aicardi-Goutières Syndrome mouse model. RAW luciferase reporter cells were exposed to dsDNA (**a**), 5'ppp-HP20 RNA (**b**), Pam3CSK4 (**c**), poly(I:C) (**d**), lipopolysaccharide (LPS) (**e**), or recombinant murine interferon-β (*Ifnb*) **f** to promote either a type I interferon (**a**), (**b**), (**f**), or NF-κB (**c**), (**d**), (**e**) response under different immune stimuli and simultaneously treated with indicated small molecule (or vehicle). Type I interferon response was read via luciferase reporter, while NF-κB response was read via qRT-PCR. **g** RAW KO-cGAS reporter cells were exposed to dsDNA or cGAMP and simultaneously treated with each of the small molecules. **h** The cytotoxic effects of the lead compounds were tested in RAW macrophages at concentrations spanning the range tested in the dose response curves. Cytotoxicity was measured by the quantitation of ATP (CellTiter-Glo assay) at 72 h and shown as percent viability of cells. **i** BMDM cells from *Trex1*$^{-/-}$ mice were treated with the indicated compound for 24 h and then harvested for qRT-PCR analysis. Shown are the results, relative to IFNB1 expression in Trex1 wild-type BMDMs. *Error bars* represent SEM

each small molecule, and compared their ability to suppress these immunogenic stimuli (Fig. 6a–f). As compared to its ability to inhibit dsDNA-dependent reporter activation, RU.521 was unable to potently suppress activation of cells by essentially all the immunogenic stimuli tested (Fig. 6a–f). These results differed from what we observed with RU.365 and RU.332. We found that RU.365 led to increased Il-6 messenger RNA (mRNA) expression in cells activated by Pam3CSK4, poly(I:C), or LPS (Fig. 6c–e); none of the compounds caused reporter activation on their own (Fig. 6a). RU.332 inhibited the activation of cells by most of the immunogenic ligands tested.

To further assess selectivity, we used RAW cells that do not express cGAS (KO-cGAS) and stimulated them with dsDNA or cGAMP. Although the introduction of dsDNA into these cells do not stimulate reporter activity given the lack of cGAS, treatment of these cells with cGAMP leads to activation via Sting stimulation. Under conditions in which the dsDNA pattern recognition pathway is stimulated by using cGAMP, we observed that RU.521 did not show any significant inhibition of reporter activity (Fig. 6g)—which is distinct from its activities in cells exposed to dsDNA when cGAS is present.

The inhibitory effects of RU.521 and RU.365 on cGAS signaling does not appear to be a consequence of cytotoxicity since we only observed ≥ 50% cellular viability loss at concentration levels significantly over their cellular IC$_{50}$ values (Fig. 6h). The least toxic compound was RU.521, where we measured an LD$_{50}$ value of 62 μM (88-fold), followed by RU.365 (45 μM, 9-fold), then RU.332 (48 μM, 3.5-fold).

**RU.521 is active in murine bone marrow-derived macrophages.** The phenotype and elevated cytokine expression levels observed in *Trex1*$^{-/-}$ knockout mice reflect the clinical presentation of some human autoimmune disorders—notably AGS and Chilblain lupus where mutations within human TREX1 render the primary cellular DNA 3′-to-5′ exonuclease non-functional[32, 33]. Previous groups have shown that Trex null mice lacking cGAS, Sting, or IRF3 expression develop normally and bear no phenotype associated with interferonopathy[35–37]. Thus, the chronically elevated levels of cytokines observed in Trex1 null mice are a consequence of constitutively activated cGAS, due to the inability to eliminate aberrantly localized self-DNA. We harvested BMDMs from 6–8-week old *Trex1*$^{-/-}$ mice, treated them with each compound, and measured expression levels of IFNB1 by quantitative reverse transcription PCR (qRT-PCR) (Fig. 6i). Treatment of primary BMDMs with RU.521 or its analogs reduced IFNB1 expression, indicating their effectiveness in suppressing intrinsic DNA-dependent, constitutively-activated type I interferon expression in cells deficient of a cytoplasmic DNA exonuclease.

**Discussion**
Using an in vitro mass-spectrometry based high-throughput compound screen, we initially identified RU.365 and its benzothiazole analog RU.332 as inhibitors of the dsDNA-induced enzymatic activity of cGAS. The generation of cGAMP by cGAS requires that the enzyme bind to dsDNA and use its two substrates, ATP and GTP, to generate the cyclic dinucleotide.

Thus, an inhibitor could potentially be found to disrupt dsDNA binding, block ATP and/or GTP from entering the active site, or somehow inhibit the generation of either phosphodiester linkage to prevent cyclization of cGAMP. Our ITC experiments demonstrated that the presence of RU.365 reduced the binding affinity of cGAS for either GTP or ATP, whereas the kinetic experiments indicated that RU.365 only affected the enzyme's $V_{max}$. These findings suggest that RU.365 is a non-competitive inhibitor against ATP or GTP. Noncompetitive inhibition is common in multi-reactant systems and can occur with active site inhibitors of multi-substrate enzymes for a variety of reasons, including: (1) the presence of exosites, (2) rate-limiting step isomerization of the catalytic site, (3) two-step mechanisms, and (4) bisubstrate/byproduct enzymes that must follow an ordered substrate binding or product release[45, 46]. In the case of cGAS, it is unlikely that an exosite exists because the substrates are not high molecular weight polymers such as DNA or protein. However, the other three mechanisms, or a combination thereof, remain a possibility.

Our previous structural studies have shown that upon dsDNA binding, cGAS is activated through conformational transitions resulting in the formation of a catalytically competent and accessible nucleotide-binding pocket for generation of cGAMP[14]. An analysis of the ternary complexes between cGAS, dsDNA and either compound indicated that the inhibitors occupy only one side of the active pocket, similar to bound cGAMP. Arg364 and Tyr421 are two highly conserved amino-acid residues found in mouse and human cGAS, and the stacking interaction mediated by these residues appears critical for inhibitor binding and also represents a conserved strategy used by different cGAS ligands (substrates/intermediate/product). Given the position of the inhibitors within the larger active pocket and its interaction with Arg364 and Tyr421, we reasoned that designing a compound that could fill the rest of the cGAS pocket and further stabilize the interaction with the two amino acids might lead to a more potent inhibitor. From a series of chemically synthesized derivatives, we subsequently identified RU.521 as exhibiting nanomolar cGAS inhibitor activity in vitro. The ternary complex with cGAS, dsDNA, and RU.521 showed that the compound had an increased stacking interaction with Arg364 and Tyr421 due to the orientation of the benzimidazole ring as well as a water-mediated interaction between RU.521, Gly290, and Lys350—which was not observed with RU.365.

Using mouse cell lines and primary cells from $Trex1^{-/-}$ mice to evaluate the lead compounds, we determined that RU.521 was the most potent inhibitor of dsDNA-induced immune activation. Using a panel of pattern recognition receptor ligands, RU.521 suppressed dsDNA-dependent activation and showed little to no inhibitory effects for the other pathways. Importantly, we demonstrated that the inhibition of the dsDNA pathway by RU.521 requires the presence of cGAS, as the small molecule did not have inhibitory activity in KO-cGAS cells under conditions in which the dsDNA sensing pathway is stimulated using cGAMP to directly activate Sting. Moreover, RU.521 did not inhibit cells directly stimulated with recombinant Ifnb, indicating that it does not appreciably target IFNB1 protein, interferon receptors, and/or downstream signaling components of the JAK/STAT pathway. Curiously, we observed that treatment of cells with RU.365, particularly in conjunction with Tlr stimulation, led to the activation of Il-6 expression. While we cannot explain the molecular basis of these RU.365 results, presumably off-target activity, none of these effects were observed with the improved compound RU.521.

Taken together, we find that RU.521 exhibited the most optimal characteristics among this class of compounds, having potent nanomolar activity in vitro and in cells, improved selectivity, and nominal cytotoxicity. As it was a compound that was specifically designed to be a more potent inhibitor of cGAS, our results underscore the value of a structurally-guided and interdisciplinary approach to rational drug design. Nonetheless, it should be noted that while we did test a reasonable number of inflammatory pathways to assess the selectivity of these small molecules, it is naturally a limited set. Further testing in whole animals and subsequent pharmacokinetic optimization will likely reveal the full extent of potential off-target effects and toxicities of RU.521. These in turn would point to where improvements can be made—and what the therapeutic index might be, which would not be an unusual course toward the eventual development of a clinically relevant derivative.

cGAS has emerged to be an essential protein for the innate immune response to cytosolic DNA given its activation of Sting by the production of cGAMP. Like Sting, cells that lack cGAS fail to substantially upregulate IFNB1 expression in response to infection by bacterial, viral, and eukaryotic pathogens[27, 47, 48], or to the accumulation of self-DNA as in the case of knockout mouse models of AGS[49–52]. Nevertheless, there are other DNA sensing proteins[5, 48], and significant questions remain as to why multiple sensors exist. Although cell-type specific expression of sensors might provide some clues, cGAS is found co-expressed in cells with other DNA sensors such as Interferon gamma inducible protein 16 (IFI16, IFI204 in *M. musculus*), which partly signals through STING[53]. While IFI16 is partially found in the cytoplasm, it is predominantly in the nucleus where it can act as a transcription co-regulator[48]. Interestingly, there are an increasing number of reports that indicate that cGAS can interact with IFI16 under various cellular and pathogenic contexts[54–56]. The underlying mechanisms that dictate the manner in which cGAS cooperates with other DNA sensors like IFI16 remains an actively researched area—one that can benefit from the use of selective cGAS inhibitors.

In summary, we have discovered and characterized a small molecule that shows potent and selective inhibition of cGAS dsDNA-dependent enzymatic activity in vitro and in cells, including primary macrophages taken from a Trex1 null mouse model of AGS autoimmunity. Based on our structural and kinetic studies, RU.521 can occupy the catalytic site of cGAS and reduce its affinity for ATP and GTP. Moreover, it is likely that RU.521 does not interfere with dsDNA directly since an acridine orange competition screen did not show any indication that it could act in a manner that could intercalate or interact with DNA. RU.521 will be a useful experimental chemical probe toward deciphering cGAS biology and with further derivatization, may prove to be an instrumental structural scaffold toward the development of an immunomodulatory therapeutic agent in treating cGAS-related human disorders.

## Methods

**Chemical libraries**. A total of 123,306 compounds were screened in the primary screen. Compounds were selected from vendor files using fingerprint-based (FCFP6) clustering algorithms with Biovia Pipeline Pilot software (http://accelrys. com/products/collaborative-science/biovia-pipeline-pilot). The average cluster size was set to 30 and cluster centers were identified and chosen for purchase. For most of the collections, PAINS were removed. However, the non-drug libraries and natural product collections contain a large number of PAINS that were permitted. Selected compounds were purchased from the following vendors: ChemDiv (7022), San Diego, CA; Spectrum (1360), New Brunswick, NJ; Prestwick (471), Illkirch, France; Enamine (53,108), Monmouth Junction, NJ; Pharmakon-900 (905), (MicroSource, Gaylordsville, CT); LifeChem (11,218), Life Chemicals, Niagara-on-the-Lake, Canada; Specs (4051), Specs, Zoetermeer, The Netherlands; Chiral Center Diversity Library (3289), NIH Clinical Collection (727); Tocris (480), Bristol, UK; Biofocus (4416), Charles River, Wilmington, MA; HTSRC Clinical Collection (294), NIH Small Molecule Repository; Cerep (640), Cerep, Poitiers, France; Chembridge (34,134), ChemBridge, San Diego, CA; AMRI (2833), AMRI, Albany, NY; Analyticon (352), AnalytiCon discovery, Potsdam, Germany; ChemX (983). One-thousand sixty-eight compounds were screened in the pilot study using Lopac

library, Sigma, Carlsbad, CA. All compounds were dissolved in DMSO at 5 mM and stored at −28 °C.

**Preparation of dsDNA for activity assays of mouse cGAS.** Non-phosphorylated oligodeoxynucleotides were purchased from IDT and full-length content was verified by separation of an aliquot on urea-containing 10% polyacrylamide gels followed by ultraviolet shadowing. Fully complementary strands were annealed at 200 μM strand concentration in buffer containing 20 mM Tris-HCl, pH 7.5, and 150 mM NaCl by first incubating at 95 °C for 90 s followed by slow cooling to 25 °C at a rate of 0.1 °C per second in a Peltier thermocycler (BioRad). Complete annealing was verified by electrophoresis using 2.5% low melting temperature agarose gel (Life Technologies, 15517-014) containing ethidium bromide and comparing the mobility with the single-stranded oligodeoxynucleotides. The sequences used for annealing can be found in Supplementary Table 5.

**High-throughput screening.** Screening reactions were carried out in 20 μl volumes in 384 small volume deep well polypropylene plates. The final concentration of cGAS enzyme, dsDNA, ATP, and GTP were 60 nM, 300 nM, 300 μM, and 300 μM, respectively. Five microliters of reaction buffer composed of 20 mM Tris-HCl pH 7.4, 150 mM NaCl, 5 mM MgCl$_2$, 1 mM dithiothreitol (DTT), and 0.01% Tween-20 were dispensed per well using a Thermo Multidrop Combi dispenser (Thermo Scientific). The liquid was collected at the well bottom using centrifugation for 30 s at 180×g. Compounds were dissolved in DMSO and 0.05 μl of 5 mM were dispensed with a Janus 384 MDT NanoHead (PerkinElmer). Final concentration of the compounds in the assay was 12.5 μM. A concentration of 0.5% DMSO did not interfere with cGAMP production from recombinant cGAS. Next, 10 μl of a master mix containing reaction buffer supplemented with 0.6 mM ATP, 0.6 mM GTP, and 0.6 μM dsDNA was added to wells in columns 1–23 using a Thermo Multidrop Combi dispenser, while 10 μl of the master mix devoid of dsDNA (control for no enzymatic activity) was added to wells in column 24. Plates were centrifuged for 30 s at 180×g to collect all liquid at the bottom of the wells. The reaction was started by adding 5 μl of 0.24 μM recombinant full-length mouse cGAS in reaction buffer to each well of the plate followed by centrifugation for 30 s at 180×g and incubation for 120 min at room temperature. The reaction was stopped by addition of 65 μl of 0.5% (v/v) formic acid per well. The plates were centrifuged for 30 s at 180×g and sealed with a Velocity11 PlateLoc thermal plate sealer. Compounds that inhibited cGAS activity by ≥ 60% were retested in concentration response experiments to determine half maximal inhibitory concentration (IC$_{50}$). Compounds were serially diluted by half for a total of ten dilutions where the highest final concentration in the assay was 25 μM. The compounds selected for follow-up studies were reordered from the vendors, dissolved in DMSO to a concentration of 10 mM and re-tested in concentration response experiments. The IC$_{50}$ values for the enzymatic assay were calculated using GraphPad Prism (7.01), from three replicate experiments; the error bars represent SD.

**RapidFire 365 mass spectrometry high-throughput assay.** All samples were analyzed using a RapidFire 365 high-throughput solid phase extraction system integrated with a 6230 TOF mass spectrometer (Agilent Technologies, Wakefield, MA, USA). The solvent used for loading/washing process was an aqueous solution of 5 mM ammonium acetate, pH 10. For elution of the analytes, the organic solvent used was 50% water, 25% acetone, and 25 % acetonitrile containing 5 mM ammonium acetate, pH 10. Each sample of approximately 35 μl were aspirated from a 384-well plate and separated using a Graphitic carbon Type D cartridge. After each sample was loaded onto the cartridge it was washed for 4 s at 1.5 ml min$^{-1}$ using the aqueous solvent. ATP, GTP and cGAMP were eluted for 5 s using the organic solvent at a flow rate of 1.5 ml min$^{-1}$. The cartridge was then re-equilibrated with the aqueous solvent for 5 s at a flow rate of 1.5 ml min$^{-1}$. All samples were analyzed using a negative ionization mode in the mass spectrometer, with a gas temperature of 350 °C, nebulizer pressure of 35 psig, gas flow rate of 15 l min$^{-1}$. The acquisition range of all chromatograms was between 300 and 800 m/z and the molecular masses of the peaks detected were the following: ATP: 505.9835, GTP: 521.9854 and cGAMP: 673.0906. The Agilent RapidFire Integrator software was used to calculate the area under the curve (AUC) of the extracted ion counts for each analyte. Data points represent values of product formation. Catalytic rates expressed as percent product formation were calculated using the AUC of cGAMP and using the AUC's of ATP and GTP to normalize for variations in the capacity of the solid phase extraction column during a screening run, as shown in the following formula: product formation (%) = [(AUC$_{cGAMP}$ × 100)/ (AUC$_{cGAMP}$ + ½ AUC$_{ATP}$ + ½ AUC$_{GTP}$)].

The EC$_{50}$ of dsDNA (six replicates) was calculated using GraphPad Prism (7.01); error bars represent SD. The time course of mouse and human cGAS assays were plotted using GraphPad Prism (7.01), with three replicates each; error bars represent SD. To determine the percent inhibition of mouse cGAS the data was normalized against the positive control (column 24) and negative control (column 23). The % inhibition is calculated as follows: % inhibition = 100 × (sample—average negative control)/(average positive control—average negative control)]. The quality of the screen was assessed by Z′ factor[57] calculated as

follows: Z′ = 1−[3∗(standard deviation positive control + standard deviation negative control)/(average positive control—average negative control)].

**Cheminformatics and data handling.** Data from all screening studies performed at The Rockefeller University are acquired and analyzed using the CDD Vault from collaborative Drug discovery (Burlingame, CA). www.collaborativedrug.com). MarvinSketch (ChemAxon, Budapest, version 14.9.8.0) was used for drawing and naming the structures. Pipeline pilot was used to search for analogs in various commercial databases.

**DNA intercalators.** Validated compounds of the primary screen were tested in a high-throughput FPassay for their capacity to intercalate DNA following the protocol developed at the Broad Institute (National Center for Biotechnology Information. PubChem BioAssay Database; AID = 504727, https://pubchem.ncbi. nlm.nih.gov/bioassay/504727 (accessed 27 April, 2016)). The assay was performed in a final volume of 30 μl in 384-solid bottom opaque plates. Ten microliters of HEN buffer (10 mM HEPES pH 7.5, 1 mM EDTA pH 7.5, 100 mM NaCl) were dispensed per well using a Thermo Multidrop Combi dispenser (Thermo Scientific). The liquid was collected at the well bottom using centrifugation for 30 s at 180×g. Compounds were dissolved in DMSO and 0.18 μl at 5 mM were dispensed with a Janus 384 MDT NanoHead (Perkin Elmer). The final concentration of compounds in the assay was 30 μM. Ten microliters of a solution of 150 nM acridine orange in HEN buffer was dispensed per well using a Thermo Multidrop Combi dispenser (Thermo Scientific). The liquid was collected at the well bottom using centrifugation for 30 s at 180×g. Subsequently, 10 μl of a solution of 45-bp dsDNA at 37.5 μg ml$^{-1}$ was dispensed per well using a Thermo Multidrop Combi dispenser (Thermo Scientific). The liquid was collected at the well bottom using centrifugation for 30 s at 180 × g and the plates were incubated for 30 min. Mitoxantrone, a known DNA intercalator was used at 50 μM as a positive control, while DMSO alone was used as negative control. FP was measured using a Biotek Synergy Neo plate reader. Samples were excited by a flash Xenon lamp filtered by a 485 nm excitation filter, collected 10 mm from the top of the well, and fluorescence emission was detected through a 530 nm filter, taking ten measurements per data point. All FP values are expressed in millipolarization (mP) units. The mP values were calculated using the equation mP = 1000 × [($I_S$−$I_{SB}$)−($I_P$−$I_{PB}$)]/[($I_S$−$I_{SB}$) + ($I_P$−$I_{PB}$)], where $I_S$ and $I_{SB}$ refer to the parallel and perpendicular emission intensity, respectively, and $I_{SB}$ and $I_{SP}$ corresponds to parallel emission intensity perpendicular emission intensity of the buffer, respectively[58]. The experimental G-factor, defined as G = $I_S$/$I_P$, was set to 1.0 in all experiments.

**Kinetics parameter of mcGAS in presence of RU.365.** Kinetics parameters of cGAMP formation were determined in the reaction mixtures carried out in 20 μl volumes in 384 small volume deep well polypropylene plates. The final concentration of m-cGAS enzyme and dsDNA were 60 nM and 300 nM, respectively. To determine if RU.365 was competitive against ATP and/or GTP, different concentrations of RU.365 ranging from 0.8 μM to 6.3 μM were tested, keeping one substrate fixed at 300 μM while varying the other one. Apparent $K_m^{app}$ and apparent $V_{max}^{app}$ values were also calculated in the presence and absence of the small molecule. RU.365 was diluted by half in DMSO for a total of four dilutions where the highest final concentration in the assay was 6.3 μM. Ten microliters of reaction buffer composed of 20 mM Tris-HCl pH 7.4, 150 mM NaCl, 5 mM MgCl$_2$, and 0.01% Tween-20 were dispensed per well using a Thermo Multidrop Combi dispenser (Thermo Scientific). The liquid was collected at the well bottom by centrifugation for 30 s at 180×g. In all, 0.1 μl of RU.365 serially diluted in DMSO were dispensed with a Janus 384 MDT NanoHead (PerkinElmer). A concentration of 0.5% DMSO did not interfere with cGAMP production from recombinant cGAS.

Each reaction containing a variable substrate concentration was performed in a different plate. A separate column containing different amounts of cGAMP was also included in each plate in order to construct the calibration curve and calculate the μmoles of product formed. Twenty different master mix solutions were prepared in reaction buffer supplemented with 2 mM DTT and 1.2 μM dsDNA. Of these, Ten master mix solutions were prepared with different concentrations of GTP ranging from 2.36 μM to 1.2 mM while keeping the concentration of ATP fixed at 1.2 mM and vice versa for the other 10 master mix solutions. Five microliters of these solutions with variable substrate concentration were added for each reaction. Finally, the reaction was started by adding 5 μl of 0.24 μM recombinant full-length mouse cGAS in reaction buffer supplemented with 2 mM DTT to each well of the plate followed by centrifugation for 30 s at 180×g and incubation for 120 min at room temperature. The reaction was stopped by addition of 60 μl of 0.5% (v/v) formic acid per well. The plates were centrifuged for 30 s at 180×g and sealed with a Velocity11 PlateLoc thermal plate sealer. To calculate IC$_{50}$ log (inhibitor) vs. response, a variable slope non-linear fit from GraphPad Prism (7.01) was used, using three replicate experiments; error bars represent SD. The kinetics parameters were obtained fitting the curves to an allosteric sigmoidal nonlinear fit using the same software.

**Isothermal titration calorimetry.** ITC measurements were performed at 25 °C using a MicroCal auto-iTC200 calorimeter (MicroCal, LLC). Purified truncated

mouse cGAS was dialyzed against 40 mM HEPES buffer (pH 7.5) and 300 mM NaCl for 24 h at 4 °C. Dialysis led to some precipitation, which were removed through centrifugation followed by collection of the supernatant. The concentration of the protein in supernatant was measured using Pierce® BCA protein assay. The protein was flash frozen in liquid $N_2$ and stored at −80 °C. For the ITC assay, the dialyzed protein was prediluted to 20 µM in the dialysis buffer and diluted to a final concentration of 10 µM in a final buffer containing 40 mM HEPES, 1% DMSO, 150 mM NaCl, 0.01% Tween-20 and +/− dsDNA 10 µM and +/− ATP or GTP 1 mM. Then, 2 µl RU.365 100 µM, dissolved in the buffer aforementioned, were injected into 0.2 ml of protein in the chamber every 150 s. Data for raw ITC and thermodynamic curves, each from one experiment, were downloaded after analysis using Affinimeter software and plotted using GraphPad Prism (7.01). The software calculates error bars as an indication of the contribution of the raw data noise that is determined from the integral of the standard deviation of the baseline during the injection.

**Synthesis of RU320521**. Please see Supplementary Fig. 8 for an illustration of the reaction scheme that was used to synthesize RU320521 (RU.521).

**2, 3-dichloro-6-nitro-aniline (2)**. A mixture of 1, 2, 3-trichloro-4-nitro-benzene (5.00 g, 22.08 mmol, 1.00 eq) and MeOH-NH$_3$ (50 ml) was heated in a steel reaction vessel at 120 °C and stirred for 24 h. The mixture was filtered. The filter cake was washed with a mixture of Petroleum ether and EtOAc (15:1, 300 ml). The solid was concentrated in vacuum, and the residue was washed with a mixture of PE: EA (15:1, 300 ml) to give 2, 3-dichloro-6-nitro-aniline (4.1 g, crude product) as yellow solid.
**1H Nuclear magnetic resonance (NMR):** (METHANOL-d$_4$, 400 MHz) δppm 8.08 (d, $J$ = 9.2 Hz, 1 H), 6.86 (d, $J$ = 9.2 Hz, 1 H).

**3, 4-dichlorobenzene-1, 2-diamine (3)**. To a solution of 2, 3-dichloro-6-nitro-aniline (2.20 g, 10.63 mmol, 1.00 eq) in MeOH (44.00 ml) was added a solution of NH$_4$Cl (568.47 mg, 10.63 mmol, 371.55 ul, 1.00 eq) in H$_2$O (2 ml). Then Zn (3.48 g, 53.15 mmol, 5.00 eq) was added slowly at 25 °C under N$_2$. The mixture was stirred at 25 °C for 2 h. The mixture was filtered through celite, and the residue was washed with EtOAc (3 × 150 ml). The combined organic layers were concentrated in vacuum to get the crude product. The crude product was purified by column chromatography (Petroleum ether: Ethyl acetate = 50:1 to 2:1) to give 3, 4-dichlorobenzene-1, 2-diamine (1.90 g, crude) as blank brown solid.
**1H NMR:** (CHLOROFORM-d, 400 MHz) δppm 6.78 (d, $J$ = 8.4 Hz, 1 H), 6.56 (d, $J$ = 8.6 Hz, 1 H), 3.91 (br s, 2 H), 3.40 (br s, 2 H).

**4, 5-dichloro-1, 3-dihydrobenzimidazol-2-one (4)**. To a solution of 3, 4-dichlorobenzene-1, 2-diamine (1.90 g, 10.73 mmol, 1.00 eq) in THF (50.00 ml) was added pyridine (1.70 g, 21.46 mmol, 1.73 ml, 2.00 eq). CDI (3.48 g, 21.46 mmol, 2.00 eq) in DCM (50.00 ml) was dropwise. The mixture was stirred at 25 °C for 12 h. The mixture was concentrated in vacuum to get crude solid. The solid was washed with Petroleum ether: Ethyl acetate = 15:1 several times to afford 4, 5-dichloro-1, 3–dihydrobenzimidazol– 2-one (1.70 g, 78.03% yield) as white solid.
**1H NMR:** (DMSO-d$_6$, 400 MHz) δppm 11.34 (s, 1 H), 11.00 (s, 1 H), 7.12 (d, $J$ = 8.2 Hz, 1 H), 6.87 (d, $J$ = 8.4 Hz, 1 H).

**2, 6, 7-trichloro-1H-benzimidazole (5)**. A flask was charged with 4, 5-dichloro-1, 3-dihydrobenzimidazol-2-one (500.00 mg, 2.46 mmol, 1.00 eq) and POCl$_3$ (24.75 g, 161.43 mmol, 15.00 ml, 65.62 eq) in one portion and was stirred at 110 °C for 12 h. Liquid chromatography-mass spectrometry (LCMS) showed one major peak with desired MS was detected. The reaction mixture was concentrated at 50 °C to afford a residue. The residue was washed with NaHCO$_3$ aqueous (50 ml) and extracted with CH$_2$Cl$_2$ (3 × 80 ml). The combined organic layers were dried with Na$_2$SO$_4$ and concentrated in vacuum to give 2, 6, 7-trichloro-1H-benzimidazole (540.00 mg, 2.44 mmol, 99.19% yield) as white solid.
**LCMS:** (M + H$^+$) : 223.0 @ 0.792 min (5–95% ACN in H$_2$O, 1.5 min).
**1H NMR:** (DMSO-d$_6$, 400 MHz) δppm 7.59–7.53 (m, 1 H), 7.53–7.48 (m, 1 H).

**(6,7-dichloro-1H-benzimidazol-2-yl)hydrazine (6)**. A flask was charged with 2, 6, 7-trichloro-1H-benzimidazole (300.00 mg, 1.35 mmol, 1.00 eq) and NH$_2$NH$_2$. H$_2$O (30.90 g, 617.27 mmol, 30.00 ml, 455.69 eq). The mixture was stirred at 100 °C for 12 h. After being cooled to ambient temperature (25 °C), water (4 ml) was added to the reaction mixture under ice cooling. The resulting precipitates were collected by filtration and washed with water (3 × 3 ml), then dried in vacuum to give (6,7- dichloro-1H- benzimidazol-2-yl)hydrazine (230.00 mg, 1.06 mmol, 78.49% yield) was obtained as a white solid.
**1H NMR:** (DMSO-d$_6$, 400 MHz) δppm 8.29 (br s, 1 H), 7.08–7.03 (m, 1 H), 7.01–6.96 (m, 1 H), 4.59 (br s, 2 H).

**3-[1-(6,7-dichloro-1H-benzimidazol-2-yl)-5-hydroxy-3-methyl-pyrazol-4-yl]- 3H-isobenzofuran-1-one (8)**. To a suspension of (6,7-dichloro-1H-benzimidazol- 2-yl)hydrazine (100.00 mg, 460.70 umol, 1.00 eq) in acetic acid (5.00 ml) was added

(ethyl 3-oxo-2-(3-oxo-1H-isobenzofuran-1-yl) butanoate (126.86 mg, 483.74 umol, 1.05 eq) in one portion at 25 °C under N$_2$. The mixture was stirred at 25 °C for 12 h, then heated to 50 °C stirred for another 3 h. The reaction mixture was diluted with water (20 ml) and extracted with EtOAc (3 × 50 ml). The combined organic layers were washed with brine (50 ml), dried over Na$_2$SO$_4$, filtered and concentrated under reduced pressure to give a residue. The residue was purified by prep-HPLC (FA) to give 3-[1-(6,7-dichloro-1H-benzimidazol-2-yl)-5-hydroxy-3-methyl–pyra zol-4-yl]-3H-isobenzofuran-1-one (37.00 mg, 19.27% yield, 99.617% purity) as a white solid.
**LCMS:** (M + H$^+$) : 415.1 @ 2.887 min (WUXIAB10, 4.5 min).
**1H NMR:** (DMSO-d$_6$, 400 MHz) δppm 7.83 (d, $J$ = 7.7 Hz, 1 H), 7.75–7.67 (m, 1 H), 7.60–7.49 (m, 2 H), 7.42–7.35 (m, 1 H), 7.33–7.27 (m, 1 H), 6.58 (s, 1 H), 2.19 (br s, 3 H).

**Protein for high-throughput screening and crystallization**. The gene encoding mouse cGAS was purchased from Open Biosystems Inc. The sequences corresponding to full-length (for high-throughput screening) and residues 147–507 (for structural study) of cGAS were inserted into a modified pRSFDuet-1 vector (Novagen), in which cGAS was separated from the preceding His$_6$-SUMO tag by an ubiquitin-like protease (ULP1) cleavage site. The gene sequences were subsequently confirmed by sequencing. The fusion proteins were expressed in BL21 (DE3) RIL bacteria (Agilent Technologies). The cells were grown at 37 °C until OD$_{600}$ reached approximately 0.6. The temperature was then shifted to 18 °C and the cells were induced by addition of isopropyl β-ᴅ-1-thiogalactopyranoside to the culture medium at a final concentration of 0.3 mM. After induction, the cells were grown overnight. The fusion protein was purified over a Ni-NTA affinity column. The His$_6$-SUMO tag was removed by ULP1 cleavage during dialysis against buffer containing 40 mM Tris-HCl pH 7.5, 0.3 M NaCl, 1 mM DTT. After dialysis, the protein sample was further fractionated over a Heparin column, followed by gel filtration on a 16/60 G200 Superdex column. The final sample of cGAS (full-length) and cGAS (147–507) contain about 30 mg ml$^{-1}$ protein, 20 mMTris-HCl pH 7.5, 300 mMNaCl, 1 mM DTT. The yield of final cGAS protein is ~ 0.7 mg (full-length) and ~ 1.2 mg (aa 147–507) from 1 l bacteria medium.

**Crystallization**. The co-crystallization approach employed for murine cGAS (147–507), in complex with dsDNA and small-molecule inhibitors (RU.365, RU.332, and RU.521), was performed largely as previously described[14]. Briefly, samples were prepared by directly mixing protein with a 16-bp DNA (containing 1-nt 5′-overhang at either end) in a 1:1.2 molar ratio, and with inhibitors in a final concentration of 1 mM. The crystals of cGAS-dsDNA-RU.365 and cGAS-dsDNA-RU.332 were generated by hanging drop vapor diffusion method at 20 °C, from drops mixed from 1 µl of cGAS-dsDNA-inhibitor solution and 1 µl of reservoir solution (0.1 M MES, pH 6.3, 26% PEG400, 0.1 M MgCl$_2$). The crystals of cGAS-dsDNA-RU.521 were generated by sitting drop vapor diffusion method at 20 °C, from drops mixed from 0.2 µl of cGAS-dsDNA-inhibitor solution and 0.2 µl of reservoir solution (0.1 M MES, pH 6.9, 22.5% PEG400, 0.08 M MgCl$_2$).

**Structure determination**. The diffraction data sets for cGAS-dsDNA-inhibitor ternary complexes were collected at the Advanced Photo Source at the Argonne National Laboratory. The diffraction data were indexed, integrated and scaled using the NECAT RAPD online server, and the structures of cGAS-dsDNA-inhibitor ternary complexes were solved using molecular replacement method in PHASER_ENREF_2[59] using our previous cGAS-dsDNA binary complex structure[14] (RCSB code: 4K96) as the search model. Model building and structural refinement was carried out using COOT[60] and PHENIX[61] software, respectively. The statistics of the data collection and refinement are shown in Supplementary Table 3.

**Cell lines**. Mouse RAW macrophages (RAW-Lucia ISG, RAW-Lucia ISG-KO-TREX1, RAW-Lucia ISG-KO-cGAS, and RAW-Blue Invivogen) were cultured in DMEM with high glucose, ʟ-glutamine, phenol red, and sodium pyruvate (Life Technologies) supplemented with 10% FBS, 100 U ml$^{-1}$ penicillin-streptomycin, Normocin (100 µg ml$^{-1}$, Invivogen), and Zeocin (200 µg ml$^{-1}$, Invitrogen) and an additional 20 mM ʟ-glutamine and 1 mM sodium pyruvate. RAW-Lucia cells stably expressed an interferon sensitive response element from the mouse Isg54 minimal promoter and five interferon-stimulated response elements (Isre-Isg54) coupled to a synthetic coelenterazine-utilizing luciferase. RAW-Blue cells have a chromosomal integration of a secreted embryonic alkaline phosphatase (SEAP) reporter construct inducible by nuclear factor-κB (NF-κB) and AP-1. Wild-type and $Trex1^{-/-}$ null derived BMDMs were maintained in RPMI 1640 with ʟ-glutamine and phenol red (Life Technologies) supplemented with 10% FBS, 10 mM HEPES, 1× non-essential amino acids, 1 mM sodium pyruvate, 0.055% 2-mercaptoethanol, 100 U ml$^{-1}$ penicillin-streptomycin, 10 ng ml$^{-1}$ of recombinant murine M-CSF (PeproTech), and an additional 2 mM ʟ-glutamine. All cell lines were routinely checked for mycoplasma contamination using MycoAlert (Lonza).

**Mice and isolation of BMDMs**. C57BL/6 wild type and $Trex1^{-/-}$ offspring from $Trex1^{+/-}$ crosses (a kind gift from Dr Dan Stetson, University of Washington) were harvested at 6–8 weeks of age. Femurs and tibias were removed from age- and

sex-matched mice. The ends were clipped and marrow pushed from bones via syringe. Clumps were broken apart via pipetting, rinsed, and pelleted in PBS, passed through a cell strainer, and plated in seven 100 mm petri dishes per mouse. Cells were matured in media described under the methods for cell lines with an additional 10 ng ml$^{-1}$ M-CSF (for a total of 20 ng ml$^{-1}$ M-CSF during the maturation period). On Day 7, cells were counted and re-plated at $5 \times 10^5$ cells ml$^{-1}$ in 100 mm petri dishes. BMDMs were maintained by adding an equal volume of media to each dish every 2 days after harvesting and performing a complete media change every 4 days after harvesting. IACUC protocols for humane termination of mice for Vanderbilt University were strictly followed.

**Cellular luciferase assays.** A concentration range of dsDNA from 0.004 to 40 µg was complexed with Lipofectamine LTX (Life Technologies) per the manufacturer's suggested protocol. The sequence and annealing conditions of the dsDNA used is described above. Typically, 0.4 µg or 0.6 µg of dsDNA was used to stimulate RAW-Lucia ISG-KO-TREX1/ISG-KO-cGAS or RAW-Lucia ISG cells, respectively, which equated to 90% of maximal luciferase activity observed. dsDNA stimulation of RAW macrophages and subsequent luciferase measurements were performed with or without the addition of small molecules at indicated concentrations in each figure, or with DMSO as vehicle control. For the dose curve experiments, serial dilutions of inhibitors were applied to $6.7 \times 10^5$ cells, plated in 24-well dishes, and co-stimulated with 0.4 µg of dsDNA. In subsequent experiments using inhibitors in RAW cells, the small molecules were used at concentrations representing the derived EC$_{75}$ values, while co-stimulated with one of the following: 0.4 µg dsDNA, 50 nM 5'ppp-HP20 hairpin RNA (a kind gift from Dr Anna Pyle, Yale University; sequence located in Supplementary Table 5), 10 µM cGAMP, or 100 U ml$^{-1}$ murine recombinant Ifnb (R&D Systems). dsDNA was transfected using Lipofectamine LTX. 5'ppp-HP20 RNA was transfected using Lipofectamine 2000. Cells were harvested 24 h post-transfection, washed with PBS, and lysed in 1× Luciferase Cell Lysis Buffer (Pierce). Luciferase luminescence was recorded with a microplate reader (BioTek Synergy HTX) using QUANTI-Luc Luciferase reagent (Invivogen) and its suggested protocol (20 µl lysate/well of a 96-well plate; microplate reader set with the following parameters: 50 µl injection, end-point measurement with 4 s start time and 0.1 s reading time).

The minimum numbers of biological and technical replicates for the luciferase assays were two and three, respectively. Analysis of luminescence values were evaluated for outliers (one standard deviation above and below the mean) for each biological replicate, and the resulting means were used to generate graphs in GraphPad Prism (v. 7.0c); where applicable, an unpaired $t$-test with Welch's correction was used to compare the results of each individual compound to dsDNA or other stimulant. The experimental variances observed within control samples (untreated or vehicle) and treated samples were similar.

**Cellular cytokine analysis of RAW macrophages.** The small molecules were used at concentrations representing the EC$_{75}$ values and were co-stimulated with either 200 ng Pam3CSK4 (Invivogen), 25 µg poly(I:C) or 100 ng lipopolysaccharide (Sigma-Aldrich) at the cell counts and conditions previously described for the cellular luciferase assays. Cells were harvested 24 h post-transfection, washed with PBS, and RNA was extracted using Trizol (Ambion) and chloroform (Sigma-Aldrich). In all, 1–2 µg of total RNA was reverse-transcribed using random hex primers and SuperScript III (Life Technologies) into cDNA. Real-time PCR was carried out with 1× FastSYBR Green Plus Master Mix (Applied Biosystems) and run on an Applied Biosystems StepOne Plus PCR machine. The expression of mRNAs repressing murine Actb1 and Il-6 were measured (see Supplementary Table 5 for primer sequences used). Target $C_T$ values were normalized to Actb1 $C_T$ values and used to calculate $\Delta C_T$. mRNA expression of target genes were then calculated using the $\Delta\Delta C_T$ method ($2^{\Delta\Delta CT}$). Expression values were analyzed as described in the method for cellular luciferase assays.

**Cytotoxicity assays.** Small-molecule compounds were serially diluted to concentrations spanning the range tested in the response curves were added to $6.7 \times 10^5$ RAW-Blue macrophages plated 16 h prior in 96-well dishes, then harvested 72 h after compound addition. ATP was measured using CellTiter Glo Viability Assay (Promega) using 50 µM Tamoxifen (Sigma) as a positive control for cytotoxicity. Viability values were generated using vehicle (DMSO) or the first dose (RU.521) as 100% and Tamoxifen as 0%. Outliers were removed as described previously.

**Cytokine expression analysis of RAW and BMDM cells.** Wild type and $Trex1^{-/-}$ BMDM cells were plated at $1 \times 10^5$ cells/well (24-well dishes) 16 h prior to addition of inhibitors at their respective IC$_{50}$ values. BMDMs were incubated in the presence of DMSO or the various inhibitors, then harvested 24 h later. RNA was extracted using Trizol and chloroform. Genomic DNA was removed using TURBO DNase (Ambion) followed by a phenol-chloroform: chloroform extraction. 480–1000 ng of total RNA was reverse-transcribed using random hex primers and SuperScript III into cDNA. Real-time PCR against murine Actb1 and IFNB1 (see Supplementary Table 5 for primer sequences used) was carried out and the data was analyzed as previously described in the method section for cytokine analysis in RAW macrophages.

**Data availability.** The pdb files representing the ternary complexes of cGAS + dsDNA + RU.365/RU.332/RU.521 have been deposited in the RCSB Protein Data Bank and can be accessed with the following codes, respectively: 5XZB, 5XZE, 5XZG. Further information regarding the high-throughput screen can be found in the Supplementary Information file. All other relevant data are available upon reasonable request.

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

## Acknowledgements

We would like to thank Dr Dan Stetson and his laboratory (University of Washington) for providing the *Trex1* null mice (originally generated by Dr Tomas Lindahl). We would like to thank the synchrotron beam line staff at the Argonne National Laboratory for their assistance. We would like to thank Dr Lauren Frick at Agilent Technologies for technical support in running the RF-MS. We thank the High Throughput and Spectroscopy Resource Center at the The Rockefeller University for providing equipment infrastructure. We also would like to thank Dr Taku Kamei (TDI) for reviewing the data on the chemical synthesis of derivatives of RU.365. Finally, we would like to thank members of the Ascano laboratory for their support, collegiality, and critical review of the manuscript. This work was supported, in part, by the following agencies: 1R35GM119569-01 (M.A.), Vanderbilt University Dept. Biochemistry start-up funds (M.A.), Rockefeller University Robertson Therapeutic Development Funds (J.F.G., T.T., and M.A.), Cancer Research Institute Irvington Postdoctoral Fellowship and NSFC31670903 (P.G.), NIGMS training grant support (4T32GM065086-14) (K.R.), and GM104962 and MSKCC core grant (P30 CA008748) (D.J.P). The crystallographic research was conducted at the Northeastern Collaborative Access Team beamlines, which are funded by NIGMS (P41 GM103403) and DOE (DE-AC02-06CH11357). The RFF-MSS instrument was purchased with funds from The Leona M. and Harry B. Helmsley Charitable Trust.

## Author contributions

J.V. contributed to the design, execution, and interpretation of all the cellular assays. C.A. contributed to the design, execution, and interpretation of the screening, biochemical, and biophysical assays, with experimental assistance from L.L., A.L., T.G., and J.S. P.G. conducted the structural biology work and the protein preparation for high-throughput screening. Y.A. was the lead chemist with R.O., T.I., and J.A. responsible for the design and preparation of the compounds based on the RU.365 scaffold. K.A. supervised the medicinal chemistry team. K.R. and S.R. ran qRT-PCR experiments and analyses. M.A., J.F.G., D.J.P., and T.T. supervised and contributed to the design and interpretation of the data. J.F.G. designed, curated, and annotated the compound library, and designed the enzymology and medicinal chemistry experiments. M.A. wrote the manuscript with editing and writing assistance from all authors.

## Additional information

**Competing interests:** The authors declare no competing financial interests.

**Change history:** A correction to this article has been published and is linked from the HTML version of this article.

