## [Peer Review file · Nature Communications]

Reviewers' comments:

Reviewer #1 (Remarks to the Author):

Ascano and co-workers describe in this manuscript identification of the first small-molecule cGAS inhibitors. cGAS plays an important role in innate immune system. Upon exposure to cytosolic DNA, cGAS produces second messenger cGAMP that activate STING to induce IFNs. As aberrant cGAS activity would lead to autoimmune diseases, specific cGAS inhibitors would potentially address this unmet medical need. The significance of this work justifies publication in Nature Communication. However, additional data is needed to support the claims that RU.365 is a useful chemical probe and provides a working scaffold for drug development.

The authors started their studies with determination of the kinetic parameters of mouse cGAS. They measured the pseudo first order rate constants (k_{cat}) and Michaelis-Menton constants (K_m) for ATP and GTP. It is not clear what enzyme concentration was used in Fig. S1. Also the expression of the rate in pmol/min is rather unusual and the volume used is not known. Without these pieces of information, K_{cat} cannot be independently derived and verified by the readers. The authors used the measurement of k_{cat} 's and K_m 's only to demonstrate the utility of RF-MS. They can easily expand this part and provide some meaningful discussions on the kinetic characteristics of cGAS. Additionally, they state that mcGAS was used because hcGAS has lower catalytic activity in vitro but no citation or supporting data is given. They should also measure the K_{cat} 's and K_m 's of hcGAS with RF-MS to provide a direct comparison.

The authors have previously done a beautiful job on X-ray analysis of cGAS. Based on their new co-crystal structures here, they argue that RU.365 and RU.332 bind to the cGAS/DNA complex and prevent the binding of both ATP and GTP. However, RU.365 only occupies half of the binding site, specifically, the ATP site. Because the pyrophosphate group of GTP somewhat clashes into RU.365 when superimposing the crystal structures of cGAS/DNA/RU.365 with cGAS/DNA/ATP/GTP, they believe that RU.365 is also GTP competitive. This is a weak argument as pyrophosphate group may swing away slightly to avoid the unfavorable interactions. After all, the cGAS/DNA/RU.365 co-crystal was obtained in the absence of ATP and GTP. It is not surprising that solvent filled the rest of the binding pocket. To support their claim, they should perform kinetic studies to demonstrate that there is competitive inhibition against both ATP and GTP. Additionally, how does the SAR data in Table S5 fit into the proposed binding mode?

The authors predicted that RU.365 can also bind to apo-cGAS as judged by the superimposed structures of cGAS/DNA/RU.365 and apo-cGAS. Later, they used MST assay to confirm the direct binding between RU.365 and apo-cGAS. Because apo-cGAS has a smaller binding pocket (is this a special case for the PDB file used or it really general?), in principle, RU.365 should bind to apo-cGAS more tightly for better hydrophobic interactions and less solvent occupancy. Can the higher affinity toward apo-cGAS be confirmed? Also the authors argue that RU.365 can inhibit cGAS at all stages. Can RU.365 compete off all or any of the ligands in Figs. 2e-g? Judging from the little correlation among the RF-MS, MST, and IFN β -Luc data in Table S3, RU.757, RU.840, and maybe RU.752 should inhibit cGAS in a different way. In particular, RU.840 showed no significant affinity toward apo-cGAS. I would like to point out that Table S3 shows very different RF-MS IC₅₀ values than those in Table S3. In Table S5, yet another IC₅₀ value was given for RU.365 for the same RF-MS assay. This is extremely confusing if not erroneous.

Regarding the SAR work, analogs RU.319 and RU.418 had no activity in RF-MS assay according to Table S5, but displayed activities comparable to those of RU.365 and Ru.332 in Fig. 5b. The large disconnection between the biochemical and cellular activities suggest that there is strong non-specific inhibition for RU.319 and RU.418. More importantly, even RU.365 and RU.332 show significant non-specific inhibition in Fig. 6 and S3. Granted that RU.365 suppressed dsDNA-stimulation more strongly, it also suppressed poly I:C, dsRNA and LPS stimulation significantly. All these graphs should be labeled with p-values to show the statistic significance. The different

expression levels of cytokines induced by RU.365 and RU.332 may attribute to difficulties in qRT-PCR analysis. However, the lack of IL-6 with RU.365 inhibition in Fig. 6c is somewhat surprising.

For Trex knockout BMDM, what is the level of IFN- β 1 as compared to that of wild-type BMDM? This information is important for validating the mouse model used in Fig. 6d. It will also be nice to see if there is a correlation with the cGAMP concentrations as well.

Reviewer #2 (Remarks to the Author):

Vincent et al report the identification and characterization of small compounds that inhibit the cytosolic DNA sensor cGAS. cGAS signals the presence of foreign or mislocalized DNA molecules in the cytosol and is a key factor in the innate immune response against a variety of viruses and bacteria. More recently, DNA sensing by cGAS has been linked to autoimmune diseases such as Aicardi-Goutieres syndrome. As such, small molecule compounds are of considerable biomedical interest.

This study reports a screening approach based on mass spectrometry and in vitro biochemical and structural characterization of compounds. Cell based studies including cells isolated from an Aicardi-Goutieres syndrome mouse model test the specificity, toxicity and efficacy.

In general, the experiments are conducted at high standards and the paper is very well written. I have no technical concerns or comments with respect to the in vitro work as well as structural studies.

A critical point in my opinion is the surprisingly low selectivity of the compounds in the cell based studies. From the data in Fig. 6a, one can argue that either cGAS is important in the cellular response to LPS, dsRNA or - perhaps more plausible - the inhibitor has considerable off-target effects. Since LPS induced activation is reduced even more than dsDNA induced activation in the case of RU.322, it is difficult to argue that the LPS effect proceeds through DNA release and cGAS. In addition, a single atom change almost swaps towards inhibition of LPS activation. Can the authors show, e.g. by using cGAMP, cGAS KO cells etc. that the observed effects for dsRNA, LPS and in BMDMs are indeed cGAS dependent? Along these lines: the title is overstated and misleading and also in the text the observed effects on selectivity appear to be exaggerated. Regarding the title, the inhibitors are neither tested in a mouse model (only in isolated cells), nor do the authors show that the reduction of interferon in these cells proceeds through cGAS as suggested in the title. Again, it would be unfortunate if the field starts to use these inhibitors to inhibit cGAS before it is clear how specific the inhibitors are.

In summary, while the overall research area is exciting and most of the results are publishable, additional data are necessary that show that the observed effects in BMDMs are cGAS dependent (perhaps using cGAMP) and whether the LPS effects are dependent on cGAS or are off-target effects. In addition, I recommend a more appropriate wording in the manuscript, in particular in the title.

Editorial comment:

The coloring of the sulfur atom in Fig. 3a and others is hard to distinguish from the color of the carbon atoms.

Reviewer #3 (Remarks to the Author):

In this manuscript by Vincent and colleagues a rational approach to identify a cGAS inhibitory molecule is described. Employing a biochemical cGAS activity assay, the authors screen a small molecule library of 123,306 compounds for mouse cGAS inhibition. Employing robust QC criteria, the authors identify a subgroup 35 compounds with considerable and presumably specific activity. Of these, 5 compounds are prioritized due to potency and chemical diversity. Subsequently, to prove defined structure activity relationships, the authors perform crystallization studies of one compound and its derivative (RU.365 and RU.332) showing that the inhibitor is bound within the active site of the enzyme. In subsequent cellular assays the authors show that the lead candidate displays considerable cGAS-inhibitory activity in both PRR-stimulated cells as well cells derived from TREX1-deficient animals.

The authors provide the description of a thoroughly performed study to identify a cGAS inhibitory molecule. However, my enthusiasm for this study is dampened by the fact that the promising inhibitory profile of the lead candidate RU.365 obtained in the in vitro studies does not translate into inhibitory activity in the cellular context (IC₅₀ of 10 μM with apparent toxicity at 10 x IC₅₀ levels). Moreover, it currently remains unclear whether this compound or other compounds identified in this study do indeed exert specific on-target activity.

Major points:

- RU.365 and more so RU.332 display considerable inhibitory activity in the context of LPS stimulation, and to a lesser extent upon poly(I:C) treatment (RU.365) and 5'-ppp-dsRNA stimulation (RU.332). Regarding the effect on TLR4 stimulation, the authors suggest that this might still be considered an on-target effect, given the fact that it has been shown that TLR4 stimulation can result in release of endogenous DNA. However, a number of reports have shown that cGAS or STING deficient cells mount normal type I IFN responses upon TLR4 stimulation (e.g., compare PMID: 27264171 or 25730264). Based on these findings it appears that RU.365 and RU.332 indeed exert considerable off-target effects in the context of unrelated PRR cascades being activated. In fact, given their structural resemblance to nucleotides, it is very well conceivable that these compounds act as kinase inhibitors (The authors could perform additional cellular assays in cGAS-deficient cells to document the suspected off-target activity).
- The authors repeatedly refer to the fact that they study a mouse model of AGS. This is in so far misleading, as the term mouse model implies the studying of live animals. The authors should rephrase this section (also the abstract) and state that they study cells from TREX1-deficient animals.
- RU.319 displays almost the same IC₅₀ as RU.332, yet it is referred to as an inactive compound following the cellular assays. This requires additional explanation. At the same time, it is not entirely clear from the data presented why RU.418 is not assigned an IC₅₀ (the other molecules also don't reach 100% inhibition).
- The toxicity studies should be performed over a broad range of compound concentrations and reported as EC₅₀ values.
- RU.755 and RU.757 display far more promising results, however, the authors do not study these compounds in details given the fact that they failed to obtain defined structure activity relationships for these molecules. Nevertheless, it would be interesting to see IC₅₀ data, EC₅₀ data and specificity data for these molecules. After all, these compounds might be most interesting molecules for subsequent studies.

Minor points:

- The RIG-I agonist should be referred to as 5' ppp-dsRNA and not 3'ppp-dsRNA.
- The authors refer to THP1 cells as human monocytes, which could be understood as primary cells. THP1 cells are a monocytic leukemia. The authors should correct this accordingly.
- When referring to recombinant interferon beta, the authors should use the term IFN- β (not IFNB1).

Reviewer #1**Major points**

“To support their claim, they should perform kinetic studies to demonstrate that there is competitive inhibition against both ATP and GTP.”

We have performed additional studies as the reviewer suggested. Using enzyme kinetics and ITC (Supplementary Figure 5 and 6) we show that while RU.365 competes for binding with substrates in ITC experiments consistent with the X-ray structure, it also shows a non-competitive mechanism of inhibition. A non-competitive mechanism of inhibition is commonly found with steady-state multi-substrate enzymes; cGAS uses the two substrates ATP and GTP. Unlike single substrate enzymes (the reviewer appeared to suggest that cGAS is a single substrate enzyme), non-competitive inhibition can be found with multi-substrate enzymes for a variety of reasons unrelated to binding pose or binding site in relation to substrate. We have revised the manuscript to reflect these new data in the results and the discussion, and also point to the following references that describe enzyme kinetics and non-classical modes of non-competitive inhibition (Segel, I. H. *Enzyme Kinetics*. Page 125, (Wiley-Interscience, 1993), and Blat, Y. *Non-competitive inhibition by active site binders*. *Chem Biol Drug Des* 75, 535–540 (2010)).

“...RU.365 and RU.332 show significant non-specific inhibition in Fig. 6 and S3. Granted that RU.365 suppressed dsDNA-stimulation more strongly, it also suppressed poly I:C, dsRNA and LPS stimulation significantly.”

We thank the reviewers for their comments particularly because it allowed us to further scrutinize the behavior of the compounds, which led to significant developments, that, we feel, has made the manuscript stronger and addresses their concerns. Namely, in the interim, we were able to make additional small molecule derivatives that were based on RU.365 and RU.332 and their interaction with cGAS. This effort led to the identification of RU.521 that we demonstrate has nanomolar inhibitory activity *in vitro* and find that it makes more significant contacts within the cGAS active pocket than its predecessors, based on the new ternary crystal data (Figures 2 and 3, and Supplementary Table 3). As written in the manuscript, the small molecule screen was conducted using recombinant mouse cGAS since the human protein exhibited significantly lower enzymatic activity at the conditions of the assay. Consequently, for consistency, we revised the manuscript to replace the THP1 experiments with a similar series of cellular assays performed in RAW macrophages (Figures 5 and 6). Using luciferase and qRT-PCR assays, we assessed the behavior of RU.365, RU.332, and the in-house synthesized derivative RU.521. Moreover, as the reviewers suggested, we used RAW macrophages that have cGAS knocked out, in order to determine whether the compounds required the presence of cGAS for their inhibitory activity. When activating cells with ligands that stimulate RIG-I or various TLRs, we found that RU.521 exhibited the most potent and selective activity, as compared to RU.365 and RU.332. We do not observe any significant inhibition of the stimulated innate pathways with the exception of the cGAS-STING pathway when cells were exposed to dsDNA. Importantly, the inhibitory effect of RU.521 required the presence of cGAS, since direct stimulation of cells with cGAMP (via STING activation) that did not express cGAS (KO-cGAS RAW cells) were refractory to RU.521 treatment. These findings were in contrast to that of the behaviors of RU.365 and RU.332.

It should be noted that while we did test a reasonable number of inflammatory pathways to assess the selectivity of the small molecules, it is naturally a limited set. Further testing in cells and in whole animals will likely reveal the full extent of RU.521 selectivity and potentially point to where improvements can be made – which would not be an unusual course of future milestones for a small molecule research program.

Minor points

“It is not clear what enzyme concentration was used...the expression of the rate in pmol/min is rather unusual and the volume used is not known. Without these pieces of information, Kcat cannot be independently derived and verified by the readers.”

Thank you for pointing out this oversight. In the paper, the rate is now expressed in $\mu\text{moles cGAMP/min}$ and the k_{cat} in min^{-1} . Moreover, it has been normalized against protein concentration and the reaction volumes are now stated in the methods (High-throughput screening and RF-MS high-throughput assay subsections). For convenience, the concentration of enzyme used was 60 nM.

“The authors used the measurement of kcat's and Km's only to demonstrate the utility of RF-MS. They can easily expand this part and provide some meaningful discussions on the kinetic characteristics of cGAS.”

Thank you for pointing this out. We have examined our data closely and conducted further experiments to show that the saturation curves consistently and significantly fit better to a sigmoidal curve rather than a hyperbolic function. This suggests that the substrate binding works cooperatively, and is consistent with structural finding of Zhang et al., *Cell*, 6, 421 (2014) who show that cGAS enzyme can exist in a 2:2 dimer with 2 enzymes binding to 2 DNA molecules. Thus it is possible that initial binding of ATP and GTP to the active site of a cGAS molecule, could cause a higher affinity binding of substrates to occur with the second cGAS molecule. This observation is stated in the text and also presented in Supplementary Figure 1. We have performed kinetic assays using RU.365; please see kinetics response in major points section.

“...mcGAS was used because hcGAS has lower catalytic activity in vitro but no citation or supporting data is given.”

We have now performed an enzyme progress curve for human, which is presented in the Supplementary Figure 2, showing that the actual signal for human cGAS is undetectable under the same conditions as mouse cGAS. Since the signal was so low, it was not suitable for accurate kinetic characterization using the technique presented in this paper.

“...in principle, RU.365 should bind to apo-cGAS more tightly for better hydrophobic interactions and less solvent occupancy. Can the higher affinity toward apo-cGAS be confirmed?”

We performed ITC experiments experiment to determine binding of RU.365 towards cGAS/dsDNA complex and apo cGAS (Supplementary Figure 4). We did not find a significant difference between the K_d of RU.365 towards cGAS/dsDNA (64.1 nM) complex or apo-cGAS (98.5 nM)

“Additionally, how does the SAR data in Table S5 fit into the proposed binding mode?”

We provide a description of how the SAR data in Supplementary Table 4 (in the revised version) fits into the proposed binding mode in the results (under “**Improving RU.365 potency through structure-activity studies**” subsection). The relevant results are discussed based on our *in vitro* assay data and our crystallization studies.

“...the authors argue that RU.365 can inhibit cGAS at all stages. Can RU.365 compete off all or any of the ligands in Figs. 2e-g?”

With this enzymatic assay we cannot demonstrate if RU.365 is competitive against the intermediate 5'-pppGpG (Fig. 2f) or 2'3'-cGAMP (Fig. 2g). Perhaps in the future we can do some crystallization trials with cGAS/dsDNA complex + RU.365 in presence of different intermediates. Also, we agree with the reviewers that the phrasing of the sentence is confusing in that it appears to suggest a claim rather than its original intent, which was to simply list the possible modes of inhibition as part of the narrative. We have re-phrased the sentences to read as “...suggests that RU.365/RU.332/RU.521 could potentially inhibit every catalytic step... To further explore this, we performed ITC binding experiments...”

“Judging from the little correlation among the RF-MS, MST, and IFN β 1-Luc data in Table S3, RU.757, RU.840, and maybe RU.752 should inhibit cGAS in a different way. In particular, RU.840 showed no significant affinity toward apo-cGAS.”

In the original version of the manuscript, we showed the data we had for RU.752, RU.757, and RU.840 since they had come out of the high-throughput screen and that we had some limited and preliminary work for these compounds. Each of these compounds represents a different chemical class from RU.365. However, for a number of reasons including compound stability (RU.755), agonistic-like behavior (RU.752 and RU.840), as well as the inability to crystallize any of these other compounds, we did not pursue them much further. Particularly given that we could not crystallize these other compounds, we felt that we would not be able to more design structurally guided derivative compounds that would ultimately improve potency and selectivity of the initial compound hit. Therefore, for the sake of clarity and to remain focused on the RU.365 series, we have removed all subsequent text and data for these other compounds in the manuscript. We now state in the results section that we prioritized our hits “based on stability, potency and structural diversity.” However, we provide additional information below since the reviewers requested further explanation.

While we did try RU.752, RU.840 and RU.757 in our interferon induction cellular assay (original Fig. 5a), for compounds RU.752 and RU.840 we found the opposite effect - meaning this compounds showed a mild activation of IFN β 1-Luc activity (even higher than the vehicle). Given these results, we decided to not pursue these compounds. RU.755 showed more promising results initially but we experienced consistent stability issues with the compound and therefore it was difficult to assess whether our preliminary results in the cells were truly from the original compound. Finally with regard to RU.757, we are investigating this compound as a different

scaffold, but it will not be included in this manuscript since the data that we have for it is still preliminary.

“...Table S3 shows very different RF-MS IC50 values than those in Table S3. In Table S5, yet another IC50 value was given for RU.365 for the same RF-MS assay.”

This was a clerical error in assembling the tables and the table has been corrected. Supplementary Table 2 contains the IC₅₀ obtained with the batch of compounds stored in the annotated library of the high throughput and spectroscopy resource center at The Rockefeller University. During the course of the studies, new batches of compounds were bought directly from the vendor and all the assays were performed with the second batch. That is the reason why the IC₅₀ values are not exactly the same since the batches are different. There was an error assembling Supplementary Tables 3 and 4; we fixed the tables in this revision.

“All these graphs should be labeled with p-values to show the statistic significance.”

The p-values for the data that show statistical significance ($p \leq 0.05$) in Fig. 6 are labeled with “*”.

“The different expression levels of cytokines induced by RU.365 and RU.332 may attribute to difficulties in qRT-PCR analysis. However, the lack of IL-6 with RU.365 inhibition in Fig. 6c is somewhat surprising.”

While we have provided a revised version of Fig.6 that has expanded the series assays into RAW cells, we also want to clarify the concern of the reviewer for the original Fig. 6c, regarding RU.365 and Il-6. RU.365 was able to inhibit dsDNA-dependent Il-6 activation; the value was so low that the bar itself was not easily discernible from the axis.

“For Trex knockout BMDM, what is the level of IFN- β 1 as compared to that of wild-type BMDM? This information is important for validating the mouse model used in Fig. 6d. It will also be nice to see if there is a correlation with the cGAMP concentrations as well.”

We apologize for this oversight. In our new Fig 6i, we provide the Ifnb1 expression levels of wild-type BMDMs. Compared to DMSO treated *Trex1*^{-/-} BMDM cells, wild-type Ifnb1 levels are ~30-fold less by qRT-PCR.

Reviewer #2**Major points**

“From the data in Fig. 6a, one can argue that either cGAS is important in the cellular response to LPS, dsRNA or - perhaps more plausible - the inhibitor has considerable off-target effects.”

We thank the reviewer for her/his comments particularly because it allowed us to further scrutinize the behavior of the compounds, which led to significant developments, that, we feel, has made the manuscript stronger and addresses their concerns. Namely, in the interim, we were able to make additional small molecule derivatives that were based on RU.365 and RU.332 and their interaction with cGAS. This effort led to the identification of RU.521 that we demonstrate has nanomolar inhibitory activity *in vitro* and find that it makes more significant contacts within the cGAS active pocket than its predecessors, based on the new ternary crystal data (Figures 2 and 3, and Supplementary Table 3). As written in the manuscript, the small molecule screen was conducted using recombinant mouse cGAS since the human protein exhibited significantly lower enzymatic activity at the conditions of the assay. Consequently, for consistency, we revised the manuscript to replace the THP1 experiments with a similar series of cellular assays performed in RAW macrophages (Figures 5 and 6). Using luciferase and qRT-PCR assays, we assessed the behavior of RU.365, RU.332, and the in-house synthesized derivative RU.521. Moreover, as the reviewers suggested, we used RAW macrophages that have cGAS knocked out, in order to determine whether the compounds required the presence of cGAS for their inhibitory activity. When activating cells with ligands that stimulate RIG-I or various TLRs, we found that RU.521 exhibited the most potent and selective activity, as compared to RU.365 and RU.332. We do not observe any significant inhibition of the stimulated innate pathways with the exception of the cGAS-STING pathway when cells were exposed to dsDNA. Importantly, the inhibitory effect of RU.521 required the presence of cGAS, since direct stimulation of cells with cGAMP (via STING activation) that did not express cGAS (KO-cGAS RAW cells) were refractory to RU.521 treatment. These findings were overall in contrast to that of the behaviors of RU.365 and RU.332.

It should be noted that while we did test a reasonable number of inflammatory pathways to assess the selectivity of the small molecules, it is naturally a limited set. Further testing in cells and in whole animals will likely reveal the full extent of RU.521 selectivity and potentially point to where improvements can be made – which would not be an unusual course of future milestones for a small molecule research program.

Minor points

“Regarding the title, the inhibitors are neither tested in a mouse model (only in isolated cells)...”

We have made several changes throughout the manuscript to make the claims more accurate; for example, we have changed the title to, **“Novel Inhibition of the DNA sensor cyclic GMP-AMP synthase reduces interferon expression in primary macrophages from an autoimmune mouse model.”** In addition, please see our response above on how our new experiments have addressed selectivity (for example through the use of cGAS KO RAW cells).

Reviewer #3**Major points**

“...it appears that RU.365 and RU.332 indeed exert considerable off-target effects in the context of unrelated PRR cascades being activated...The authors could perform additional cellular assays in cGAS-deficient cells to document the suspected off-target activity.”

We thank the reviewer for her/his comments particularly because it allowed us to further scrutinize the behavior of the compounds, which led to significant developments, that, we feel, has made the manuscript stronger and addresses their concerns. Namely, in the interim, we were able to make additional small molecule derivatives that were based on RU.365 and RU.332 and their interaction with cGAS. This effort led to the identification of RU.521 that we demonstrate has nanomolar inhibitory activity *in vitro* and find that it makes more significant contacts within the cGAS active pocket than its predecessors, based on the new ternary crystal data (Figures 2 and 3, and Supplementary Table 3). As written in the manuscript, the small molecule screen was conducted using recombinant mouse cGAS since the human protein exhibited significantly lower enzymatic activity at the conditions of the assay. Consequently, for consistency, we revised the manuscript to replace the THP1 experiments with a similar series of cellular assays performed in RAW macrophages (Figures 5 and 6). Using luciferase and qRT-PCR assays, we assessed the behavior of RU.365, RU.332, and the in-house synthesized derivative RU.521. Moreover, as the reviewers suggested, we used RAW macrophages that have cGAS knocked out, in order to determine whether the compounds required the presence of cGAS for their inhibitory activity. When activating cells with ligands that stimulate RIG-I or various TLRs, we found that RU.521 exhibited the most potent and selective activity, as compared to RU.365 and RU.332. We do not observe any significant inhibition of the stimulated innate pathways with the exception of the cGAS-STING pathway when cells were exposed to dsDNA. Importantly, the inhibitory effect of RU.521 required the presence of cGAS, since direct stimulation of cells with cGAMP (via STING activation) that did not express cGAS (KO-cGAS RAW cells) were refractory to RU.521 treatment. These findings were overall in contrast to that of the behaviors of RU.365 and RU.332.

It should be noted that while we did test a reasonable number of inflammatory pathways to assess the selectivity of the small molecules, it is naturally a limited set. Further testing in cells and in whole animals will likely reveal the full extent of RU.521 selectivity and potentially point to where improvements can be made – which would not be an unusual course of future milestones for a small molecule research program.

“The authors should rephrase this section (also the abstract) and state that they study cells from TREX1-deficient animals”

We have made several changes throughout the manuscript to make the claims more accurate; for example, we have changed the title to, **“Novel Inhibition of the DNA sensor cyclic GMP-AMP synthase reduces interferon expression in primary macrophages from an autoimmune mouse model.”**

“RU.319 displays almost the same IC50 as RU.332, yet it is referred to as an inactive compound following the cellular assays. This requires additional explanation.”

We have done a more exhaustive SAR experimental series on analogs of RU.365 and summarize the results in the new Supplementary Tables 4 and 5. The results from these experiments led us to identify RU.521 as having potent inhibitory activity as compared to the IC₅₀ values for RU.319 or RU.418 (both >25 μM, see Supplementary Table 5). It would not have been feasible to perform cellular assays for all the compounds tested in the SAR tests, nor would it have made sense to include compounds that exhibited high micromolar inhibitor activity *in vitro*, and therefore we have removed the panels containing the concentration response curves for these two compounds.

“The toxicity studies should be performed over a broad range of compound concentrations and reported as EC50 values.”

We have re-performed the toxicity assays over a broad range of compound concentrations and present this data in Figure 6g.

“RU.755 and RU.757 display far more promising results, however, the authors do not study these compounds in details given the fact that they failed to obtain defined structure activity relationships for these molecules.”

In the original version of the manuscript, we showed the data we had for RU.752, RU.755, RU.757, and RU.840 since they had come out of the high-throughput screen and that we had some limited and preliminary work for these compounds. Each of these compounds represents a different chemical class from RU.365. However, for a number of reasons including compound stability (RU.755), agonistic-like behavior (RU.752 and RU.840), as well as the inability to crystallize any of these other compounds, we did not pursue them much further. Particularly given that we could not crystallize these other compounds, we felt that we would not be able to more design structurally guided derivative compounds that would ultimately improve potency and selectivity of the initial compound hit. Therefore, for the sake of clarity and to remain focused on the RU.365 series, we have removed all subsequent text and data for these other compounds in the manuscript. We now state in the results section that we prioritized our hits “based on stability, potency and structural diversity.” However, we provide additional information below since the reviewers requested further explanation.

RU.755 showed promising results initially but we experienced consistent stability issues with the compound and therefore it was difficult to assess whether our results in the cells were truly from the original compound. Finally with regard to RU.757, we are investigating this compound as a different scaffold, but it will not be included in this manuscript since the data that we have for it is still preliminary.

Minor points

“The RIG-I agonist should be referred to as 5' ppp-dsRNA and not 3'ppp-dsRNA.”

We have corrected the typographical error by referring to the RIG-I agonist as 5'ppp-HP20, which is the RNA hairpin RIG-I ligand provided by Dr. Anna Pyle's laboratory (Kohlway et al. EMBO Rep. 14, 772-779 (2013))

“The authors refer to THP1 cells as human monocytes...”

In the revised version of our manuscript, we have replaced the THP1 data (and text) and instead show a series of experiments using RAW macrophages and BMDM primary cells.

“When referring to recombinant interferon beta, the authors should use the term IFN- β (not IFN β 1).”

We now refer to recombinant mouse interferon beta as Ifn- β in the text.

Reviewers' comments:

Reviewer #1 (Remarks to the Author):

This is a significantly improved manuscript with a good amount of new data. The authors have addressed most of the issues previously raised by the reviewers. The only remaining concern is the interpretation of the kinetic data. As one key point of this paper is that RU.365 occupies the cGAS active site, the author should comment on the specific reason (instead of simply "variety of reasons") why non-competitive kinetics is observed in their multi-substrate system. The cases in the cited reference seem to be different from the current one.

Also the authors stated that "RU.365 occupies only one side of the cGAS pocket, as does cGAMP in our previously reported structure." Do they actually mean ATP instead of cGAMP? cGAMP appears to occupy the entire pocket.

It is somewhat strange that, given the subtle structural difference, RU.332 and RU.365 modulate the immune response to stimuli in a dramatically different way (Fig. 6). Similarly, is there any reason why RU.521, with simply an introduction of two chlorine atoms to RU.365, has no off-target effects?

Reviewer #2 (Remarks to the Author):

The authors satisfactorily addressed my points and I find the manuscript suitable for publication

Reviewer #4 (Remarks to the Author):

In the submitted manuscript Dr. Ascano and colleagues describe and characterize small-molecule inhibitors of the intracellular DNA sensor cGAS. A high-throughput in vitro assay based on murine (!) cGAS and MS was used to interrogate ~ 120.000 compounds. Using stringent selection criteria 4 compounds were identified that blocked the enzymatic activity of cGAS. Following up on one compound, termed RU.365, the investigators determine crystal structures of RU.365 and cGAS and perform complementary in vitro assays, which lead them to conclude that RU.365 directly interacts with cGAS and blocks cGAMP synthesis in a non-competitive manner. Moreover, structure-activity relationship assays result in the development of more potent derivatives of RU.365, including RU.521. The possible translation application of these findings is provided by experiments using murine macrophages that are either stimulated with dsDNA (cGAS) and various controls or that have constitutive activation of the cGAS pathway due to defects in TREX1. Specifically, it is shown that pretreatment of cells with RU-521 blocks cGAS-controlled cytokine induction. The findings described in this investigation would be of interest to an audience interested in basic aspects of cGAS biology.

While the experiments related to the first part of the manuscript including the structural characterization are straightforward, the relevance of the identified small molecules to selectively inhibit cGAS in a cellular system is less convincing. First, it appears that some of the identified compounds including RU.365 and RU.332 are not specific against cGAS – in particular RU.332 seems to severely affect innate pathways in general (e.g., Figure 6b-g). Given this, dose-titration experiments in cells treated with distinct innate immune ligands are crucial to assess whether or not the other inhibitors, most importantly RU.521, can be used to reliably target cGAS in cells. Without these controls, it would be impossible to conclude specificity with regards to the compound`s inhibitory capacity in a cellular system. Although the investigators do some effort by performing dose-titrations performed in the context of the cell viability, this assay does not go far enough to rule out side-effects on other, cGAS-unrelated pathways. One additional point to consider – though not essential – applies to usage of an IFN β 1 reporter system. Although I do see

the point of the usage of a reporter system, a validation of compound RU.521 on type I IFN expression via qPCR or protein measurements (e.g., ELISA) could further substantiate and generalize the applicability of the findings.

Related to the applicability of the findings, all assays were performed on murine versions of cGAS in vitro and in cells. How similar are murine and human cGAS? Or in other words does the most promising compound RU.521 similarly work in human cells or on human cGAS? It should be feasible to test the small-molecule RU-521 in human cells in a similar manner as presented in Figure 6 and with appropriate dose-titration and toxicity assays. Without that piece of data it remains questionable whether the findings can be useful for patients.

Reviewer #1
Major points

“The only remaining concern is the interpretation of the kinetic data.”

In particular regard to the models put forth by reference #46 (*Chem Biol Drug Des* 2010; 75:535-540. Non-competitive Inhibition by Active Site Binders), a small molecule can bind within the active site of multi-substrate enzymes and exhibit non-competitive inhibitory characteristics if the enzyme undergoes one of a number of specific reaction mechanisms. We do not favor a model in which cGAS contains an exosite since the precedence for such mechanisms (e.g. prothrombinase complex) have large molecular-weight substrates like other proteins. Instead it remains plausible that the cGAS reaction mechanism requires the isomerization of its catalytic site (as it transitions from catalyzing the first phosphodiester linkage to the second and final cyclization step), a two-step mechanism, or a bisubstrate mechanism that requires an ordered substrate binding or product release. These three possibilities are not necessarily distinct since a two-step mechanism could require two substrates (e.g. ATP and GTP) for the sequential generation of two phosphodiester linkages – which may require active site isomerization between the two phosphodiester-linking reactions. Indeed the exact reaction mechanism that generates the asymmetric cyclic dinucleotide remains unknown, although previous reports (*Cell* (2013) 153(5):1094-1107, and *Cell Reports* (2015) 59:891-903) have speculated that the linear dinucleotide intermediate that forms prior to cyclization may need to leave the catalytic pocket so that it can be re-oriented for the second phosphodiester bond to be generated.

The explanation above is more briefly summarized in an amendment of the first paragraph of the Discussion section; below is an excerpt of the text:

“...Noncompetitive inhibition is common in multi-reactant systems and can occur with active site inhibitors of multi-substrate enzymes for a variety of reasons, including: 1) the presence of exosites, 2) rate-limiting step isomerization of the catalytic site, 3) two-step mechanisms, and 4) bisubstrate/byproduct enzymes that must follow an ordered substrate binding or product release^{45,46}. In the case of cGAS, it is unlikely that an exosite exists because the substrates are not high molecular weight polymers such as DNA or protein. However, the other three mechanisms, or a combination thereof, remain a possibility.”

Reviewer #4**Major points**

“While the experiments related to the first part of the manuscript including the structural characterization are straightforward, the relevance of the identified small molecules to selectively inhibit cGAS in a cellular system is less convincing...”

We believe that our revised manuscript had addressed these concerns since they were similar to that of reviewer #2 and the original reviewer #3. That said, we appreciate reviewer #4's comments since it pointed out that we needed to revise the manuscript text to better clarify our assessment of compound selectivity and why RU.521 emerged to be the improved cGAS inhibitory compound.

First, as the reviewer acknowledges, RU.365 was the first compound identified in the primary high-throughput screen with RU.332 being found as a structural analog. We were able to crystallize cGAS with these compounds and, as a result, be able to rationally design and synthesize a series of small molecules that was predicted to bind tighter to cGAS, within its active pocket. RU.521 represents these efforts. RU.521 exhibits nanomolar activity *in vitro* and makes more significant contacts within the cGAS active pocket than its predecessors, based on the new ternary crystal data (*Figures 2 and 3, and Supplementary Table 3*). Using luciferase and qRT-PCR assays, we assessed the behavior of RU.365, RU.332, and the in-house synthesized derivative RU.521. When activating cells with ligands that stimulate RIG-I or various TLRs, we found that RU.521 exhibited the most potent and selective activity, as compared to RU.365 and RU.332.

We would further like to point out that while we did not perform an exhaustive dose response curve for every single non-DNA immunogenic ligand, we did perform a dose response curve for each compound in cells in Figure 5, under dsDNA-stimulated conditions. These experiments then allowed us to use concentrations that represent each compound's cellular EC₇₅ (or IC₇₅) – to determine if the higher inhibitory concentration could lead to off-target effects (*please see the Cellular luciferase assays in the Methods section*). Under these conditions, we did not observe any significant inhibition of the stimulated innate pathways by RU.521, with the exception being the cGAS-STING pathway when cells were exposed to dsDNA. **Importantly, we also used RAW macrophages that have cGAS knocked out, in order to determine whether the compounds required the presence of cGAS for their inhibitory activity.** The inhibitory effect of RU.521 required the presence of cGAS, since direct stimulation of cells with cGAMP (via STING activation) that did not express cGAS (KO-cGAS RAW cells) were refractory to RU.521 treatment. These findings were overall in contrast to that of the behaviors of RU.365 and RU.332. We do not claim to move further with RU.365 or RU.332; they have served their purpose towards the development of RU.521. While there is some value in testing RU.521 by a full dose response against other non-DNA innate immune ligands, we see greater value in further optimizing the molecule in order to increase its potency thereby reducing its potential off-target effects, as has been an effective small molecule medicinal chemistry strategy for many compounds that eventually yield clinically applicable derivatives. At its present state, the concentrations used (EC₅₀ to EC₇₅) for RU.521 did not demonstrate observable off-target effects.

In order to address the reviewer's concerns more clearly and to also temper our conclusions regarding selectivity, we revised our *Discussion* section to include the following text in the third-to-last paragraph. [Redacted]

“...As [RU.521] was a compound that was specifically designed to be a more potent inhibitor of cGAS, our results underscore the value of a structurally-guided and interdisciplinary approach to rational drug design. Nonetheless, it should be noted that while we did test a reasonable number of inflammatory pathways to assess the selectivity of these small molecules, it is naturally a limited set. Further testing in whole animals and subsequent pharmacokinetic optimization will likely reveal any potential off-target effects and toxicities of RU.521. These in turn would point to where improvements can be made – and what the therapeutic index might be, which would not be an unusual course towards the eventual development of a clinically relevant derivative.”

“...does the most promising compound RU.521 similarly work in human cells or on human cGAS?”

As written in the manuscript (please see *Discovery of a novel low molecular weight cGAS inhibitor* section of Results), the small molecule screen was conducted using recombinant mouse cGAS since the human protein exhibited significantly lower enzymatic activity at the conditions of the assay. To support this claim, we performed an enzyme progress curve for human, which is presented in the Supplementary Figure 2. It shows that the signal for human cGAS is undetectable under the same conditions as mouse cGAS. Since the signal was so low, it was not suitable for accurate kinetic characterization using the technique presented in this paper. For consistency, we used murine macrophage cells (cell lines or primary culture, Figures 5 and 6). Throughout the manuscript, starting with the abstract, we make no claims that our compounds are essentially ‘ready’ and useful in a clinical setting. As even the reviewer also acknowledges, what we claim is that our compound “*would be of interest to an audience interested in basic aspects of cGAS biology*” – that it can be a useful chemical probe to dissect mechanism. That said, we do recognize and state that our compounds may prove to be valuable molecular scaffolds that future clinically potent derivatives could be based, but do not go beyond that speculation towards its direct applicability to patients. At its present state, we feel that this compound will already be useful in evaluating mouse models of disease that implicate DNA-sensing based immunological responses, and pave the way for compound testing that can inhibit mouse and human cGAS – which is one of our long term objectives.

[Redacted]

[Redacted]

Minor points

“...a validation of compound RU.521 on type I IFN expression via qPCR or protein measurements (e.g., ELISA) could further substantiate and generalize the applicability of the findings.”

We would like to point out that Figures 6c, 6d, 6e, and 6i were all done by real-time PCR analysis of Il-6 or Ifnb1 mRNA. *[Redacted]*